# The reciprocal relationships between social media self-control failure, mindfulness and wellbeing: A longitudinal study

Jie Du[ID]*, Peter Kerkhof, Guido M. van Koningsbruggen

Department of Communication Science, Vrije Universiteit Amsterdam, Amsterdam, The Netherlands

* dujie520@hotmail.com

## Abstract

This paper aims to shed light on the question whether, and how, social media self-control failure is related to mindfulness and wellbeing. Using a 3-wave longitudinal design, the present study among 594 daily social media users examined the reciprocal relationships between social media self-control failure and mindfulness, and between social media self-control failure and wellbeing (as assessed by subjective vitality and life satisfaction). Results of the random-intercept cross-lagged panel model showed that social media self-control failure has a time-invariant negative association with mindfulness and subjective vitality. No full reciprocal influence was found between social media self-control failure and mindfulness, yet part of this trajectory was observed, suggesting that social media self-control failure could impair mindfulness, which, in turn, might increase future social media self-control failure. For wellbeing, life satisfaction was found to predict subsequent drops in social media self-control failure.

**Data Availability Statement:** The anonymized raw datasets and syntaxes for data analysis are available online at https://osf.io/hzy8r/.

## Introduction

As social media use (e.g., Facebook, Twitter, WhatsApp) is pervasive in people's daily lives, many people have become "permanently online and permanently connected" to the online world [1]. Using social media can benefit various gratifications such as social contact and entertainment [2, 3]. However, several studies have also pointed out social media use as a behavior that may disturb achieving everyday personal goals such as getting enough sleep, or devoting sufficient time to working, studying, doing house duties or engaging in sports [3–5]. Importantly, when goal-conflict occurs, social media users frequently fail to persist with these goals and turn to social media use, even though they are aware of the possible negative outcomes [6]. Social media self-control failure is often related to procrastinatory behaviors, but may also occur in situations which are not considered as typical procrastination situations (e.g., unintended checking social media while driving a car). Social media self-control is related to more general, trait-like self-control which may affect behaviors across different situations, yet focuses on the specific self-control failure induced by social media, which may happen repeatedly and frequently in a media-rich environment. Recent studies showed that social

**Funding:** This work was supported by a grant from the China Scholarship Council (https://www.csc.edu.cn/) awarded to JD. The funders had no role in study design, data collection and analysis, decision to publish, or preparation of the manuscript.

**Competing interests:** The authors have declared that no competing interests exist.

media induced self-control failure may account for 35% of the time people spent on social media [7], making them delay other important tasks, or use their time less efficiently [8, 9]. This indicates that being unable to control one's social media use when needed might be a prevalent problem.

The prevalence of social media self-control failure has raised concerns that frequent failure in controlling social media use may be detrimental to users' cognitive control processes. We propose that such failure may be associated with decreased mindfulness—a general quality of consciousness which serves to maintain and sustain attention to the present experience, and to disengage from automatic thoughts, habits, and unhealthy behavior patterns [10, 11]. The recent surge in attention for mindfulness has been attributed to the rise of social media and mobile phones, and the continuous distractions they provide (e.g., [12]). Previous studies have examined the association between mindfulness and Internet use [13], problematic smartphone use [14], Facebook use [15], and online vigilance [16]. However, whether impaired mindfulness also emerges from multiple daily instances of social media self-control failure that many social media users experience is not yet known.

In addition, failure in controlling one's social media use could induce negative emotions due to postponing other goals [5, 6]. When using social media delays other tasks, people experience feelings of guilt about their media use [7], as well as strain and time pressure [9]. This in turn, might lead to decrease in wellbeing [6]. While existing studies supported the negative associations between problematic media use and wellbeing, little is known about the direction of this association. Uncertainty remains about possible relationships over time that may indicate their causal relationship. The same can be said for social media self-control failure and mindfulness. Therefore, using a three-wave longitudinal design, the study aimed to examine whether and how social media self-control failure is related to mindfulness and wellbeing.

Besides, differences in social media self-control failure, mindfulness and wellbeing consist of both stable individual differences between persons (traits) and change over time within persons (states). Without disentangling these two source of variance, one may draw spurious conclusions about whether their longitudinal relationship is characterized by within-person change or characterized by more stable between-person differences [17]. Thus, a growing body of media research aims to distinguish between-person differences and within-person change (e.g., [18, 19]). In line with these recent research practices in media research and given the nature of the psychological constructs we aimed to study, we used a random-intercept cross-lagged panel model (RI-CLPM) [20] to analyze the longitudinal associations between social media self-control failure, mindfulness and wellbeing. We believe this may provide a more appropriate causal picture of how these constructs are related over time.

## Social media self-control failure and mindfulness

The concept of mindfulness stems from a form of meditation practice in Buddhism [21]. It is often defined as the disposition or state of being attentive to the ongoing activities or experiences in a nonjudgmental and receptive way [10, 11, 21, 22]. One of the most important components of mindfulness is the capability of acting with awareness. For instance, Bishop et al. (2014) posit that mindfulness is a two-component construct which consists of the self-regulation of attention and adopting an orientation toward one's experiences in the present moment [22]. Baer et al. (2016) identified five components of mindfulness, two of which (i.e., observing one's inner experience and acting with awareness) reflect the component of self-regulation of attention in Bishop's definition [23]. Acting with awareness characterizes mindfulness as a state in which one can sustain and maintain attention to ongoing events and experiences. Mindfulness can be seen as an enhanced state of consciousness [21] which allows the observing

of one's internal or external state, that is, to "stand back and simply witness the current experience, rather than be immersed in it" [21]. Moreover, mindfulness increases feelings of vividness and awareness of one's current experience, which helps to clarify one's needs or interests [11], and to disengage from automatic thoughts and mind wandering [10].

Mindfulness promotes self-control through directing one's attention in a deliberate manner [24]. Self-control is known as the ability to override automatic behavioral tendencies towards temptations that conflict with higher standards or goals [6, 25]. During this process, mindfulness could serve as a 'detector' of the affective cues related to these higher standards or goals [26]. For example, people with better mindfulness might be more sensitive to the affective cues related to guilty feelings when other immediate eating desire conflicts with their dieting goals. Conversely, people with lower mindfulness might be less sensitive in detecting these affective cues, which makes it more difficult in deploying self-control against the temptations. Repeated failing to respond to affective cues, thereby experiencing self-control failure, might further decrease people's sensitivity to those cues over time. In other words, self-control failure is associated with lower mindfulness. Empirical evidence supported this idea, showing that better self-control in general was related to higher levels of mindfulness (e.g., [27, 28]).

Regarding social media-induced self-control failure, people who fail more often to control their social media use might also show less mindfulness in their daily lives. A notable example is distraction by instant messaging [29] or notifications [30], which for many people occurs numerous times each day (an average of 64 times per day according to Pielot, Church, and De Oliveira [31]). The easy accessibility of social media creates social pressure to be always available online [32] and generates over-concern about what happens online [33]. Social media thus creates difficulties for people to maintain sufficient awareness of ongoing activities or goals. Besides, everyday use of social media is often characterized by habitual social media checking behavior, which promotes automatic use of social media [34], and which further weakens the involvement of consciousness. Together, the evidence suggests that social media self-control failure might be related to the capability to deploy attention and awareness to the present tasks or goals, which is one significant component of mindfulness.

Importantly, the association between social media self-control failure and mindfulness may be bidirectional. On the one hand, frequent failure in controlling social media use leads to a constantly deprived tendency to be attentive to and aware of ongoing tasks or goals. On the other hand, being unable to be attentive to and aware of ongoing tasks also increases the likelihood to be distracted by random notifications, irrelevant thoughts, and automatic checking habit of social media. Indeed, there is evidence indicating that social media use is problematic for people who showed lower levels of mindfulness. For example, a recent study found that the frequency of texting during driving was related to lower trait mindfulness through decreased self-control, which increased the risk of car accidents [35]. Another study showed that employees' level of Facebook use was associated with lower mindfulness and subsequent burnout [15], which suggests that mindfulness may act as a buffer against the negative consequences of social media use. Conversely, lower mindfulness may also predict social media self-control failure. For example, one study found that increased mindfulness was associated with less problematic smartphone overuse [36]. Another study showed that trait mindfulness significantly decreased problematic phone use during driving [37]. Furthermore, Calvete, Gámez-Guadix, and Cortazar showed that two specific aspects of mindfulness (i.e., the extent to which one is able to observe and act with awareness) predicted deficient self-regulation of Internet use among adolescents six months later [13]. These findings suggest that lower mindfulness may indicate a higher probability of media-induced self-control failure.

Although the evidence above indicates possible mutual influences between social media self-control failure and mindfulness, no study has yet examined their mutual influences over

time. Thus, the possible causal relationship between the two variables remains unknown. From a dynamic perspective, in the long term, social media self-control failure might predict lower mindfulness, which in turn, might predict a higher probability of social media self-control failure in the future, and vice versa. Such reciprocal effects have been the focus of several recent studies in communication research. For instance, Baumgartner et al. (2017) found that adolescents' media-related multitasking predicted their later attention problems, which in turn, predicted future media multitasking [18]. Boer, Stevens, Finkenauer, and Van den Eijnden (2020) found that social media use problems were reciprocally related to adolescents' attention deficit hyperactivity disorder-symptom [38]. In a similar way, we aimed to explore the reciprocal relationships between social media self-control failure and mindfulness.

As stated before, social media self-control failure, mindfulness and wellbeing contain both enduring trait-like individual differences, and state-like within-person variance indicating change over time. For example, a recent study found that the probability people fail to control their social media varies within persons, but the likelihood of momentary failure also differs from person to person [39]. Moreover, Du et al. (2018) found that social media self-control failure was only moderately correlated with trait-like self-control (i.e., trait self-control and depletion sensitivity) [7], indicating that it may also capture state-like variance. The same can be said for mindfulness. At the between level, people's trait-like, dispositional mindfulness may differ from person to person; at the within-level, their mindfulness performance can vary across time, as shown in previous research which used mindfulness training to improve people's mindfulness performance [10]. This suggests that when exploring the reciprocal relationships of the two constructs, disentangling the trait-like individual differences and within-person change over time is necessary. To do so, we applied a random-intercept cross-lagged panel model to study the reciprocal relationships between social media self-control failure and mindfulness. Building upon the above evidence, we hypothesized that[a]:

H1: Cross-sectionally, social media self-control failure and mindfulness are negatively related.

H2: Longitudinally, social media self-control failure and mindfulness are negatively and reciprocally related.

## Social media self-control failure and wellbeing

Wellbeing is often defined as the presence of positive emotions, low levels of negative emotions, and the positive judgment of one's life in general [6, 40]. It is well-established that self-control plays an important role in wellbeing. This is because self-control prevents people from being distracted by pleasurable but unwanted desires, which benefits the goal-pursuit process [41]. Previous research has shown that people with higher trait self-control reported higher momentary happiness even as they experience the pleasurable desires [42]. This might be due to the fact that self-control over unwanted desire per se might promote the initiation of goal-pursuit behaviors [43]. This practice could lead to a less conflicting life experience, which benefits general life satisfaction. Conversely, people with lower self-control could experience that they have more conflict to deal with, and more emotional distress related to this conflict, which negatively affects their wellbeing.

Similar reasoning can be applied to self-control failure induced by social media. People who fail to control social media use are typically aware that they do so [6], which might lead to feelings of guilt or shame about their social media use [7], and in the end negatively affect their wellbeing. For example, one study found that using Facebook induced procrastination and increased students' academic stress and strain [9]. Insufficient self-control over Internet use

was found to lead to stress, anxiety and depression, particularly for those with a higher level of trait procrastination [44]. Moreover, insufficient self-control over (social) media use may also result in diminished physical activity [45] and undermine sleep quality [46], both of which are related to lower wellbeing. Overall, the evidence above implies that social media self-control failure might have negative implications for different domains of wellbeing.

However, whether decreased wellbeing conversely leads to more self-control failure in social media use is less clear. Theoretically, low levels of wellbeing might increase the risk of social media self-control failure because people may seek ways to alleviate their negative emotions. As social media can be used for mood regulation (e.g., entertainment, social contact) [47], using social media could become an attractive way to alleviate a bad mood, which challenges one's self-control. Moreover, decreased wellbeing might also result from stress generated by goals related to study and work, because they typically require volitional effort and delaying them can create time pressure [6, 9]. The attempt to release stress thus could make social media temptation become even stronger.

The evidence above suggests a reciprocal relationship between social media self-control failure and wellbeing. Regarding the possible reciprocal relationship between social media self-control failure and wellbeing, recent studies mainly focused on the frequency of social media use and its relationship with wellbeing. The findings were inconsistent. For instance, research based on a large-scale panel dataset showed that social media use was negatively and reciprocally related over time to teenagers' wellbeing, although the effect sizes were trivial [48]. Another study examined the reciprocal association between use of social network sites and wellbeing. Results showed that at the between-person level, people who more often use social network sites reported lower wellbeing. However, at the within-person level, no reciprocal association was found between social network sites use and wellbeing [49]. Moreover, a meta-analysis of 124 studies showed that only certain types of social media use (e.g., online gaming) negatively impact wellbeing, while other media use (e.g., texting) positively affects wellbeing [50].

An important reason for the inconsistent findings could be that social media use per se does not necessarily result in negative outcomes, whereas *problematic* social media use may account for the negative outcomes, because people can use social media intensively without disturbing other daily goals. For instance, a meta-analysis showed that whereas overall social media use had no significant association with adolescents' wellbeing, there was a significant negative association between deficient self-control over social media use and wellbeing [51]. Also, a recent longitudinal study examined the reciprocal relationship between social media use intensity and wellbeing, as well as the reciprocal relationship between problematic social media use and wellbeing. Results showed that higher levels of problematic social media use, rather than social media use intensity, were consistently related to lower life satisfaction one year later. But reversely, life satisfaction did not predict more problematic social media use [52].

Social media self-control failure occurs when the desire to use social media conflicts with other important goals, turning social media use into a temptation. This is distinct from social media desires in a goal-compatible situation, such as willingly using social media to be entertained after work. This stresses that frequently occurred social media self-control failure is a problematic form of social media, which could harm wellbeing. Based on the evidence above, as well as the potential reciprocal path from wellbeing to social media self-control failure, we propose the following hypothesis and research question:

H3: Cross-sectionally, social media self-control failure and wellbeing are negatively related.

RQ: Do social media self-control failure and wellbeing reciprocally and negatively influence each other?

## Method

The research questions, hypotheses, design and sample size of the study were pre-registered before data collection (https://aspredicted.org/pb5ig.pdf). The anonymized raw datasets and syntaxes for data analysis is available online at https://osf.io/hzy8r/. This study meets all requirements of standard research code of conduct of the Faculty of Social Sciences at Vrije Universiteit Amsterdam (https://fsw.vu.nl/en/research/research-ethics-review/index.aspx) and no further ethical evaluation is required.

### Participants and procedure

The study is part of a three-wave longitudinal research with an interval of 4 months. Besides the measures in the current study, in the first wave, we also assessed several factors related to the predictors of social media self-control failure (the results of which have been reported in a previous paper [2])[b]. We did not include these measures in this paper because the aim of the present study was to examine the reciprocal effects of social media self-control failure on mindfulness and wellbeing. Participants were recruited online through the Prolific online participant pool (https://prolific.ac).

Participants were initially asked to provide consent for their participation. Then they were instructed to complete questionnaires about their social media use, social media self-control failure, mindfulness, and our measures of wellbeing (subjective vitality and life satisfaction). Demographic information was collected at the end of the survey. Four months after the T1 survey, those who participated in the T1 survey were invited by email to take part in the T2 survey, in which the same questionnaires were used to assess their social media use, mindfulness, subjective vitality and life satisfaction. This was repeated at T3. The median completion time for the survey in T1 to T3 was 391s, 211s and 197s respectively[c].

The present study recruited social media users aged from 16 to 60. After excluding participants that did not meet the pre-registered exclusion criteria (see Table 1), the total sample of the study was 594: 594 participants completed the T1 survey, 410 (69%) the T2 survey and 329 (55%) the T3 survey. In total, 270 (45%) participants completed all three waves of the survey and 465 (79%) participated in at least two waves of the survey. Participants had an average age of 33.8 ($SD$ = 9.78), 73% were female participants. Participants were mostly from the United Kingdom (87%), mostly employed for wages (52%), mostly had the highest educational level of undergraduate (36%), and their mostly selected daily used social media platforms were Facebook (86%) (see Table 2 for more detailed information). Kruskal-Wallis test showed that

**Table 1. Data exclusion criteria.**

|  | Time1 | Time2 | Time3 |
|---|---|---|---|
| (1) Partially filled-out survey by the same participant (duplicates)[a]. | 8 | 8 | 5 |
| (2) Age did not range between 16 and 60. | 26 | — | — |
| (3) Participants who did not use social media at least once a day. | 0 | 10 | 8 |
| (4) Participants who did not complete the survey that caused missing values in the main variables of the model[a]. | 0 | 4 | 1 |

[a] Criterion (1) and (4) were not pre-registered.

**Table 2. Descriptive statistics of participants.**

| Country of residence (%) $N$ = 594 | | Education (%) $N$ = 594 | | Mostly selected daily used social media platforms (%) $N$ = 594 | | | |
|---|---|---|---|---|---|---|---|
| United Kingdom | 86.5 | Undergraduate degree (BA/BSc/other) | 36.4 | Facebook | 86.4 | Snapchat | 21.9 |
| United States | 7.7 | College/A levels | 29.3 | Whatsapp | 58.2 | Pinterest | 14.8 |
| Canada | 3.9 | Secondary school/GCSE | 16.0 | Facebook Messenger | 53.9 | LinkedIn | 8.4 |
| | | Graduate degree (MA/MSc/MPhil/other) | 13.5 | Youtube | 53.4 | Google+ | 7.1 |
| Other | 1.2 | Doctorate degree (PhD/MD/other) | 3.0 | Instagram | 46.5 | Skype | 6.4 |
| Australia | 0.7 | No formal qualifications | 1.2 | Twitter | 31.8 | Other[a] | < 5 |
| | | Other | 0.7 | | | | |

[a] Tumblr (4.4), Other (3.9), Viver (3.2), Vine (0.3), Wechat (0.8), MySpace (0.2), QQ/Qzone/I do not use social media (0)

participants' time spent on social media differed between the three waves ($\chi^2$ = 10.38, $df$ = 2, $p$ = .01). Post-hoc tests (Dunn's test) using the Bonferroni correction showed that participants in T1 reported a higher social media usage than in T2 ($M_{diff}$ = 2.77, $p$ = .008) and T3 ($M_{diff}$ = 2.58, $p$ = .015). No difference was found in social media usage between the T2 and T3 surveys. Their self-reported frequency of social media use also differed between the three waves ($\chi^2$ = 6.48, $df$ = 2, $p$ = .04). No pairwise difference was found in social media frequency between the T1, T2 and T3 surveys.

## Measurements

To assess social media self-control failure, the social media self-control failure (SMSCF)-scale [7] was used. Participants were asked "How often do you give in to a desire to use social media even though your social media use at that particular moment: 1)...conflicts with other goals (for example: doing things for school/study/work or other tasks)? 2)...makes you use your time less efficiently? 3)...makes you delay other things you want or need to do?" Each item was rated on a 5-point scale (1 = Never, 5 = Always).

To measure daily experienced mindfulness, we used the Mindful Attention Awareness Scale (MAAS, e.g., "I could be experiencing some emotion and not be conscious of it until some time later;" "I find it difficult to stay focused on what's happening in the present") [21]. The MAAS specifically addresses day-to-day circumstances of mindfulness. Each item was rated on a 6-point scale (1 = Almost always, 6 = Almost never). A higher score indicates more mindfulness.

For measuring wellbeing, we used two widely assessed wellbeing concepts: 1) subjective vitality; 2) life satisfaction. Subjective vitality reflects the positive feeling of aliveness and energy, which is identified as a dynamic reflection of wellbeing [53]. To measure perceived subjective vitality, we used the Subjective Vitality Scale (SVS, e.g., "I feel alive and vital;" "I have energy and spirit"; 1 = Not at all true, 5 = Very true) [53].

Life satisfaction is described as the overall judgment of one's quality of life compared with his/her expectations [40]. To assess participants' overall wellbeing, we used the Satisfaction With Life Scale (SWLS, e.g., "In most ways my life is close to my ideal;" "The conditions of my life are excellent"; 1 = Strongly disagree, 7 = Strongly agree) [40].

All scales in the model showed good internal reliability and a unidimensional structure (see Tables 3 and 4 for scale descriptives and correlation matrix). We conducted a confirmatory factor analysis to check the construct validity for each of the scale at each wave. Results showed that a one-factor model with all items had a good fit for each of the scales (See S1 File for full results). For each scale, the items were averaged into a mean index. The missing value pattern of the scales was completely at random (MCAR) across all waves according to Little's MCAR

**Table 3. Descriptive statistics of measures in all waves.**

| | SMSCFt1 | SMSCFt2 | SMSCFt3 | MFt1 | MFt2 | MFt3 | SVt1 | SVt2 | SVt3 | LSt1 | LSt2 | LSt3 | Visitt1 | Visitt2 | Visitt3 | Uset1 | Uset2 | Uset3 |
|---|---|---|---|---|---|---|---|---|---|---|---|---|---|---|---|---|---|---|
| N | 594 | 410 | 329 | 594 | 410 | 329 | 594 | 410 | 329 | 594 | 410 | 329 | 594 | 410 | 329 | 594 | 410 | 329 |
| $M/Mdn^a$ | 3.08 | 3.09 | 3.00 | 3.84 | 3.81 | 3.82 | 2.94 | 2.80 | 2.78 | 4.16 | 4.22 | 4.25 | 4.00 | 4.00 | 4.00 | 4.00 | 3.00 | 3.00 |
| $SD/Mode^b$ | 1.02 | 0.97 | 0.96 | 0.83 | 0.83 | 0.82 | 0.91 | 0.85 | 0.87 | 1.46 | 1.42 | 1.40 | 3.00 | 3.00 | 3.00 | 4.00 | 2.00 | 2.00 |
| Min | 1.00 | 1.00 | 1.00 | 1.00 | 1.60 | 1.47 | 1.00 | 1.00 | 1.00 | 1.00 | 1.00 | 1.00 | 1.00 | 1.00 | 1.00 | 2.00 | 2.00 | 2.00 |
| Max | 5.00 | 5.00 | 5.00 | 6.00 | 6.00 | 6.00 | 5.00 | 5.00 | 4.86 | 7.00 | 7.00 | 7.00 | 6.00 | 6.00 | 6.00 | 6.00 | 6.00 | 6.00 |
| ω | .905 | .886 | .894 | .901 | .915 | .913 | .917 | .914 | .928 | .919 | .924 | .929 | — | — | — | — | — | — |
| Eigenvalue Variance (%) | 2.52 (83.8) | 2.44 (81.4) | 2.47 (82.4) | 6.34 (42.3) | 6.91 (46.1) | 6.81 (45.4) | 4.67 (66.6) | 4.61 (65.8) | 4.88 (69.7) | 3.77 (75.3) | 3.83 (76.5) | 3.88 (77.7) | — | — | — | — | — | — |

SMSCF = social media self-control failure, MF = mindfulness, SV = subjective vitality, LS = life satisfaction, Visit = frequency of visit on social media, Use = time consumption of social media.

[a]Median scores were reported for frequency of visit and time consumption of social media.

[b]Mode scores were reported for frequency of visit and time consumption of social media.

**Table 4. Correlation matrix of the variables in the model across three waves.**

|  | SMSCFt1 | SMSCFt2 | SMSCFt3 | MFt1 | MFt2 | MFt3 | SVt1 | SVt2 | SVt3 | LSt1 | LSt2 | LSt3 |
|---|---|---|---|---|---|---|---|---|---|---|---|---|
| SMSCFt1 | — | | | | | | | | | | | |
| SMSCFt2 | 0.63*** | — | | | | | | | | | | |
| SMSCFt3 | 0.66*** | 0.72*** | — | | | | | | | | | |
| MFt1 | -0.42*** | -0.40*** | -0.42*** | — | | | | | | | | |
| MFt2 | -0.43*** | -0.48*** | -0.52*** | 0.78*** | — | | | | | | | |
| MFt3 | -0.38*** | -0.45*** | -0.48*** | 0.74*** | 0.82*** | — | | | | | | |
| SVt1 | -0.19*** | -0.21*** | -0.12* | 0.29*** | 0.35*** | 0.21*** | — | | | | | |
| SVt2 | -0.19*** | -0.28*** | -0.22*** | 0.34*** | 0.43*** | 0.34*** | 0.79*** | — | | | | |
| SVt3 | -0.20*** | -0.26*** | -0.21*** | 0.25*** | 0.42*** | 0.33*** | 0.78*** | 0.77*** | — | | | |
| LSt1 | -0.08* | -0.14** | -0.08 | 0.22*** | 0.20*** | 0.15** | 0.60*** | 0.54*** | 0.53*** | — | | |
| LSt2 | -0.11* | -0.19*** | -0.19** | 0.25*** | 0.31*** | 0.31*** | 0.52*** | 0.60*** | 0.53*** | 0.84*** | — | |
| LSt3 | -0.09 | -0.16** | -0.11* | 0.19*** | 0.27*** | 0.25*** | 0.48*** | 0.53*** | 0.56*** | 0.78*** | 0.84*** | — |

$N_{T1}$ = 594, $N_{T2}$ = 410, $N_{T3}$ = 329, SMSCF = social media self-control failure, MF = mindfulness, SV = subjective vitality, LS = life satisfaction.

* $p < .05$,

** $p < .01$,

*** $p < .001$.

test (SMSCF: $\chi^2$ = 17.104, $df$ = 15, $p$ = .313; MAAS: $\chi^2$ = 50.150, $df$ = 74, $p$ = .985; SVS: $\chi^2$ = 26.163, $df$ = 35, $p$ = .860; SWLS: $\chi^2$ = 23.085, $df$ = 25, $p$ = .573).

Moreover, we assessed participants' social media use with two questions: 1) "On average, approximately how many minutes per day do you spend on social media?" (1 = 10 *minutes or less*, 6 = *3+ hours*), 2) "On average, how often do you visit social media?" (1 = *Less than once a day*, 6 = *More than 3 times an hour*). Spearman's correlations between the two questions were: $r_{T1}$ = 0.604, $r_{T2}$ = 0.620, $r_{T3}$ = 0.621, all $p$s < .001.

## Design and data analysis

We used the random intercept cross-lagged panel model (RI-CLPM) to test the reciprocal influences between people's social media self-control failure, mindfulness and wellbeing [20]. The RI-CLPM is often applied in examining bidirectional effects of longitudinal panel data. Compared with the traditional cross-lagged panel model, the assumption of the RI-CLPM is that psychological constructs (e.g., behavior, emotion, cognition) may change within a person across time while taking into account stable individual differences within a group [20]. The model posits that the bidirectional influence of psychological constructs over time is composed by the within-individual changes of each construct over time (the autoregressive paths), the within-individual reciprocal influence of these constructs over time (the cross-lagged paths), and the more stable, time invariant between-person differences of these constructs (the random intercepts). Thus, instead of taking the psychological constructs as an overall measure varying around their temporary group means, the RI-CLPM disentangles the variance of the between-person differences of the constructs across time [20].

As indicated above, the missing value pattern was completely at random (MCAR) across all waves for scales used in the model (Little's MCAR test: $\chi^2$ = 115.28, $df$ = 150, $p$ = .984). The multivariate normality assumption across all variables of the model was rejected (Mardia Skewness = 145816.02, Mardia Kurtosis = 23.73). Therefore, we employed the full-information maximum likelihood (FIML, [54]) and bootstrap estimator (bootstrap = 5000) to estimate the model.

## Results

We tested three RI-CLPM models for mindfulness, subjective vitality and life satisfaction respectively. For each model, we first tested a model with no constraints for the autoregressive paths and the cross-lagged paths ($df = 1$, Model A). Considering the same time interval between each wave (4 months), individual change between each wave can be treated as equivalent, therefore we constrained the autoregressive paths to be equal ($df = 3$, Model B). We also tested a model in which the cross-lagged paths were constrained to be equal ($df = 3$, Model C), since one could also assume that the bidirectional influences between SMSCF and the outcomes are steady across time based on the same time interval. Finally, we constrained both kinds of lags to be equal ($df = 5$, Model D). We compared the fit of the unconstrained models with models in different constraint conditions to see if constraints increase the model fit (see Table 5). Results showed that no differences were found between models in different constraint conditions, except for the model for subjective vitality, in which constraining the autoregressive paths significantly decreased the model fit compared with the unconstrained model. This means that constraining the autoregressive paths and/or cross-lagged paths to be equal does not increase the model fit (while for subjective vitality, constraining the autoregressive paths decreases the model fit). Considering that each model has different assumptions which might affect the conclusions regarding the reciprocal relationships, we report the results for all the constraint conditions.

### Reciprocal model for social media self-control failure and mindfulness

The models under all constraint conditions showed good fit (see Table 5). Fig 1 shows the reciprocal model of social media self-control failure and mindfulness in three waves as an example of the RI-CLPM. Tables 6–8 presents the results of the reciprocal model regarding social media self-control failure and mindfulness (Table 6), subjective vitality (Table 7), and life satisfaction (Table 8), under the four different constraint conditions.

At the within-person level, a partial reciprocal effect was found between social media self-control failure and mindfulness in both the unconstrained model and the model which constrained the autoregressive paths to be equal: SMSCF in T1 indicated lower mindfulness in T2, and mindfulness in T2 indicated more social media self-control failure in T3. However, mindfulness in T1 did not indicate more SMSCF in T2, which in turn did not predict mindfulness in T3. No reciprocal relationships were found in the two other models.

Intra-wave residual correlations of social media self-control failure and mindfulness in T2 and T3 were negative and significant in all constraint conditions, indicating that when an individual scored higher than his/her expected scored on social media self-control failure, he/she scored lower than expected on mindfulness. The intra-wave correlation in T1 was significant in conditions that only constrained autoregressive paths to be equal and constrained both autoregressive and cross-lagged paths to be equal.

At the between-person level, social media self-control failure and mindfulness were negatively and significantly correlated in all constraint conditions, meaning that regardless of within-person change, the trait-like differences between individuals in SMSCF and mindfulness were negatively correlated.

In order to check for the robustness of the results, in an exploratory analysis we added age, gender, and social media use (both the frequency of visit on social media and time spent on social media) as control variables to the model. The control variables were selected based on the findings in other studies, which showed that age was positively related to social media self-control failure, social media use was negatively related to social media self-control failure, and female participants tend to report higher levels of social media self-control failure than male

**Table 5. Model comparisons for different constraint conditions.**

| Model | | $\chi^2$ | df | p | CFI | TLI | SRMR | RMSEA [90% CI] | Model comparison | $\Delta\chi^2$ | $\Delta df$ | $\Delta p$ |
|---|---|---|---|---|---|---|---|---|---|---|---|---|
| SMSCF-Mindfulness | A1 | 0.151 | 1 | 0.697 | 1.000 | 1.009 | 0.003 | 0.000 [0.000, 0.080] | | | | |
| | B1 | 2.628 | 3 | 0.453 | 1.000 | 1.001 | 0.014 | 0.000 [0.000, 0.066] | A1 vs. B1 | 2.477 | 2 | 0.290 |
| | C1 | 3.760 | 3 | 0.289 | 0.999 | 0.997 | 0.017 | 0.021 [0.000, 0.075] | A1 vs. C1 | 3.137 | 2 | 0.208 |
| | D1 | 5.765 | 5 | 0.330 | 0.999 | 0.998 | 0.023 | 0.016 [0.000, 0.061] | A1 vs. D1 | 5.614 | 4 | 0.230 |
| SMSCF-Subjective vitality | A2 | 3.045 | 1 | 0.081 | 0.998 | 0.976 | 0.014 | 0.059 [0.000, 0.139] | | | | |
| | B2 | 9.482 | 3 | 0.024 | 0.995 | 0.975 | 0.027 | 0.060 [0.020, 0.105] | A2 vs. B2 | 6.437 | 2 | 0.040* |
| | C2 | 5.444 | 3 | 0.142 | 0.998 | 0.990 | 0.018 | 0.037 [0.000, 0.086] | A2 vs. C2 | 2.399 | 2 | 0.301 |
| | D2 | 10.252 | 5 | 0.068 | 0.996 | 0.988 | 0.026 | 0.042 [0.000, 0.079] | A2 vs. D2 | 7.206 | 4 | 0.125 |
| SMSCF- Life satisfaction | A3 | 0.021 | 1 | 0.884 | 1.000 | 1.010 | 0.001 | 0.000 [0.000, 0.054] | | | | |
| | B3 | 0.253 | 3 | 0.969 | 1.000 | 1.010 | 0.005 | 0.000 [0.000, 0.000] | A3 vs. B3 | 0.231 | 2 | 0.891 |
| | C3 | 0.078 | 3 | 0.994 | 1.000 | 1.010 | 0.002 | 0.000 [0.000, 0.000] | A3 vs. C3 | 0.057 | 2 | 0.972 |
| | D3 | 0.418 | 5 | 0.995 | 1.000 | 1.010 | 0.006 | 0.000 [0.000, 0.000] | A3 vs. D3 | 0.397 | 4 | 0.983 |

SMSCF = Social media self-control failure, Model A: unconstrained model, Model B: constrained the autoregressive paths to be equal, Model C: constrained the cross-lagged paths to be equal, Model D: constrained both autoregressive paths and cross-lagged paths to be equal.

*$p < .05$

participants [2, 7]. Specifically, the control variables were set to predict the observed variables in three waves, and its effect on each set of variables (e.g., mindfulness from time1 to time 3, social media self-control failure from time 1 to time 3) was constrained to be equal at each occasion (see Mulder & Hamaker (2020) for more information regarding the practice of including a time-invariant predictor in RI-CLPM) [55]. The full results can be found in S2 File.

Regarding the associations of the trait-like individual differences between the two variables, the results remained unchanged after controlling for age, gender and time spent on social media. Yet, adding frequency of visiting social media to the model did affect the results which

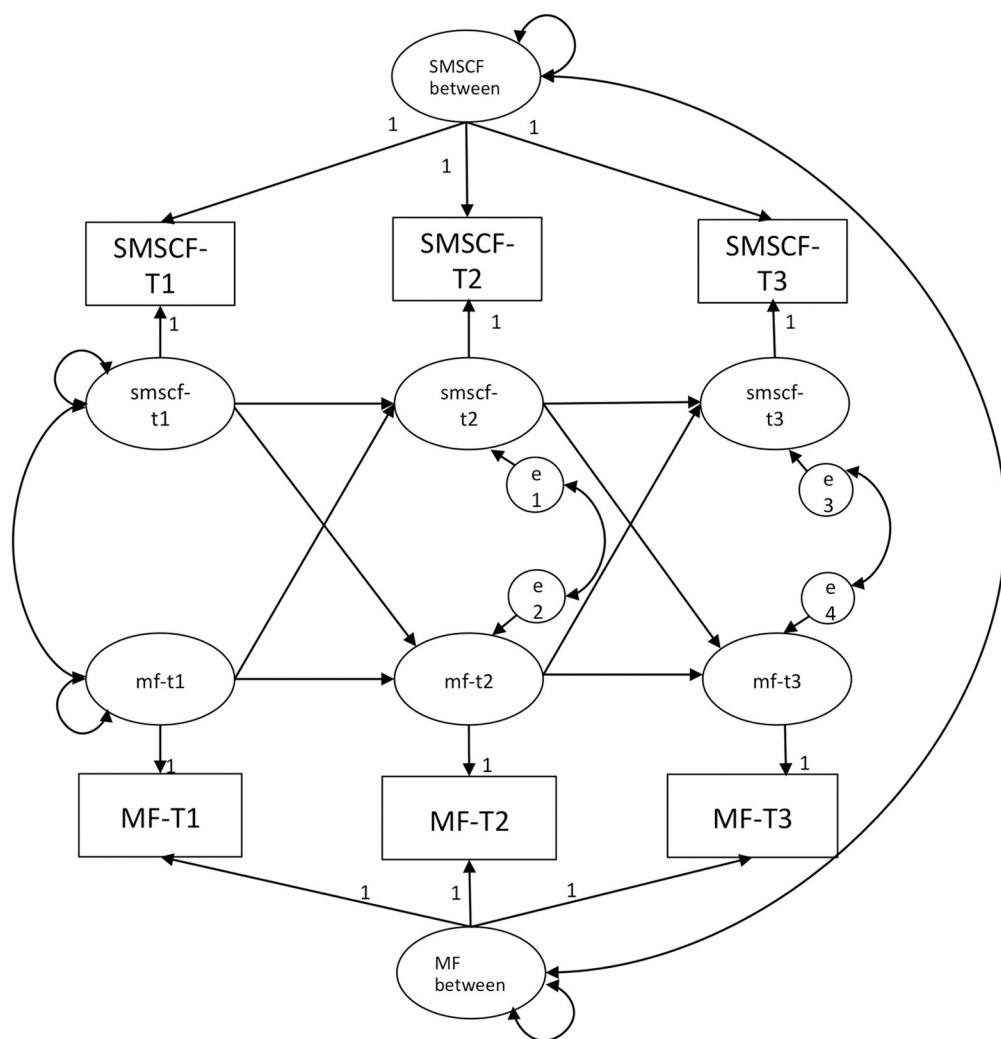

**Fig 1. Random intercept cross-lagged panel model of social media self-control failure and mindfulness.** smscf/
SMSCF = Social media self-control failure, mf/MF = Mindfulness.

in one the four models (the models which constrained the cross-lagged and autoregressive
paths) were no longer significant.

Regarding the within-person change of the two variables, adding age resulted in a full recip-
rocal effect between social media self-control failure and mindfulness in the model which con-
strained both autoregressive and cross-lagged paths to be equal. Social media self-control
failure predicted lower mindfulness and vice versa. Yet, after controlling for gender or fre-
quency of visit, the prediction from social media self-control failure T1 to mindfulness T2
were no longer significant in the unconstraint model. After controlling for time spent on social
media, the cross-lagged paths were no longer significant.

## Reciprocal model for social media self-control failure and subjective vitality

The models under all constraint conditions showed acceptable fit (see Table 5). No cross-
lagged effects were found in any of the constraint conditions. The intra-wave correlation was

**Table 6. Reciprocal relationships between social media self-control failure and mindfulness.**

| | Unconstrained model (*df* = 1) | | | Constrained autoregressive paths (*df* = 3) | | | Constrained cross-lagged paths (*df* = 3) | | | Constrained autoregressive and cross-lagged paths (*df* = 5) | | |
|---|---|---|---|---|---|---|---|---|---|---|---|---|
| | *b* | *SE* | *p* | *b* | *SE* | *p* | *b* | *SE* | *p* | *b* | *SE* | *p* |
| Autoregressive paths | | | | | | | | | | | | |
| SMSCF T1 → SMSCF T2 | -0.075 | 0.101 | 0.456 | -0.057 | 0.092 | 0.533 | -0.110 | 0.105 | 0.295 | -0.059 | 0.101 | 0.558 |
| SMSCF T2 → SMSCF T3 | -0.017 | 0.127 | 0.896 | -0.057 | 0.092 | 0.533 | 0.034 | 0.126 | 0.791 | -0.059 | 0.101 | 0.558 |
| MF T1 → MF T2 | 0.238 | 0.200 | 0.232 | 0.393 | 0.173 | **0.023*** | 0.286 | 0.202 | 0.157 | 0.377 | 0.166 | **0.022*** |
| MF T2 → MF T3 | 0.424 | 0.128 | **0.001**** | 0.393 | 0.173 | **0.023*** | 0.388 | 0.143 | **0.007**** | 0.377 | 0.166 | **0.022*** |
| Cross-lagged paths | | | | | | | | | | | | |
| SMSCF T1 → MF T2 | -0.148 | 0.075 | **0.049*** | -0.148 | 0.062 | **0.018*** | -0.096 | 0.059 | 0.107 | -0.113 | 0.065 | 0.075 |
| SMSCF T2 → MF T3 | -0.016 | 0.089 | 0.861 | -0.058 | 0.105 | 0.557 | -0.096 | 0.059 | 0.107 | -0.113 | 0.065 | 0.075 |
| MF T1 → SMSCF T2 | -0.078 | 0.213 | 0.716 | -0.191 | 0.185 | 0.302 | -0.258 | 0.164 | 0.115 | -0.314 | 0.176 | 0.082 |
| MF T2 → SMSCF T3 | -0.435 | 0.181 | **0.017*** | -0.444 | 0.186 | **0.017*** | -0.258 | 0.164 | 0.115 | -0.314 | 0.176 | 0.082 |
| Covariance | | | | | | | | | | | | |
| Between-person | -0.286 | 0.046 | **0.000***** | -0.269 | 0.069 | **0.000***** | -0.283 | 0.324 | **0.004**** | -0.270 | 0.050 | **0.000***** |
| Intra-wave correlation T1 | -0.066 | 0.040 | 0.100 | -0.080 | 0.063 | 0.205 | -0.067 | 0.323 | 0.836 | -0.081 | 0.039 | **0.040*** |
| Intra-wave correlation T2 | -0.085 | 0.037 | **0.023*** | -0.101 | 0.040 | **0.011*** | -0.087 | 0.040 | **0.030*** | -0.099 | 0.044 | **0.026*** |
| Intra-wave correlation T3 | -0.047 | 0.020 | **0.018*** | -0.049 | 0.020 | **0.016*** | -0.043 | 0.019 | **0.010*** | -0.044 | 0.020 | **0.026*** |

$N_{T1}$ = 594, $N_{T2}$ = 410, $N_{T3}$ = 329, SMSCF = Social media self-control failure, MF = Mindfulness.

***$p$ < .001,

**$p$ < .01,

*$p$ < .05

**Table 7. Reciprocal relationships between social media self-control failure and subjective vitality.**

| | Unconstrained model (*df* = 1) | | | Constrained autoregressive paths (*df* = 3) | | | Constrained cross-lagged paths (*df* = 3) | | | Constrained autoregressive and cross-lagged paths (*df* = 5) | | |
|---|---|---|---|---|---|---|---|---|---|---|---|---|
| | *b* | *SE* | *p* | *b* | *SE* | *p* | *b* | *SE* | *p* | *b* | *SE* | *p* |
| Autoregressive paths | | | | | | | | | | | | |
| SMSCF T1 → SMSCF T2 | -0.167 | 0.109 | 0.126 | -0.143 | 0.098 | 0.144 | -0.188 | 0.106 | 0.077 | -0.145 | 0.104 | 0.152 |
| SMSCF T2 → SMSCF T3 | -0.045 | 0.173 | 0.797 | -0.143 | 0.098 | 0.144 | -0.043 | 0.159 | 0.786 | -0.145 | 0.104 | 0.152 |
| SV T1 → SV T2 | 0.041 | 0.109 | 0.707 | -0.056 | 0.140 | 0.688 | 0.030 | 0.109 | 0.782 | -0.027 | 0.130 | 0.835 |
| SV T2 → SV T3 | -0.510 | 0.418 | 0.222 | -0.056 | 0.140 | 0.688 | -0.363 | 0.392 | 0.355 | -0.027 | 0.130 | 0.835 |
| Cross-lagged paths | | | | | | | | | | | | |
| SMSCF T1 → SV T2 | 0.016 | 0.070 | 0.823 | -0.030 | 0.093 | 0.743 | -0.036 | 0.061 | 0.561 | -0.055 | 0.059 | 0.350 |
| SMSCF T2 → SV T3 | -0.267 | 0.231 | 0.248 | -0.103 | 0.119 | 0.395 | -0.036 | 0.061 | 0.561 | -0.055 | 0.059 | 0.350 |
| SV T1 → SMSCF T2 | -0.194 | 0.147 | 0.187 | -0.155 | 0.161 | 0.384 | -0.128 | 0.140 | 0.360 | -0.212 | 0.132 | 0.110 |
| SV T2 → SMSCF T3 | -0.195 | 0.302 | 0.519 | -0.319 | 0.223 | 0.153 | -0.128 | 0.140 | 0.360 | -0.212 | 0.132 | 0.110 |
| Covariance | | | | | | | | | | | | |
| Between-person | -0.155 | 0.037 | **0.000***** | -0.152 | 0.037 | **0.000***** | -0.160 | 0.037 | **0.000***** | -0.148 | 0.037 | **0.000***** |
| Intra-wave correlation T1 | -0.018 | 0.026 | 0.491 | -0.020 | 0.032 | 0.533 | -0.035 | 0.020 | 0.080 | 0.036 | 0.021 | 0.089 |
| Intra-wave correlation T2 | -0.064 | 0.035 | 0.069 | -0.067 | 0.033 | **0.044*** | -0.046 | 0.038 | 0.235 | -0.067 | 0.036 | 0.065 |
| Intra-wave correlation T3 | -0.039 | 0.036 | 0.286 | -0.037 | 0.032 | 0.241 | -0.018 | 0.024 | 0.437 | -0.021 | 0.023 | 0.378 |

$N_{T1}$ = 594, $N_{T2}$ = 410, $N_{T3}$ = 329, SMSCF = Social media self-control failure, SV = Subjective vitality.

***$p$ < .001,

**$p$ < .01,

*$p$ < .05

**Table 8. Reciprocal relationships between social media self-control failure and life satisfaction.**

| | Unconstrained model (*df* = 1) | | | Constrained autoregressive paths (*df* = 3) | | | Constrained cross-lagged paths (*df* = 3) | | | Constrained autoregressive and cross-lagged paths (*df* = 5) | | |
|---|---|---|---|---|---|---|---|---|---|---|---|---|
| | *b* | *SE* | *p* | *b* | *SE* | *p* | *b* | *SE* | *p* | *b* | *SE* | *p* |
| Autoregressive paths | | | | | | | | | | | | |
| SMSCF T1 → SMSCF T2 | -0.119 | 0.102 | 0.243 | -0.107 | 0.091 | 0.239 | -0.122 | 0.099 | 0.217 | -0.106 | 0.092 | 0.251 |
| SMSCF T2 → SMSCF T3 | -0.078 | 0.171 | 0.649 | -0.107 | 0.091 | 0.239 | -0.060 | 0.147 | 0.683 | -0.106 | 0.092 | 0.251 |
| LS T1 → LS T2 | 0.213 | 0.172 | 0.214 | 0.185 | 0.153 | 0.226 | 0.224 | 0.167 | 0.181 | -0.131 | 0.082 | 0.108 |
| LS T2 → LS T3 | 0.143 | 0.233 | 0.538 | 0.185 | 0.153 | 0.226 | 0.164 | 0.175 | 0.348 | -0.131 | 0.082 | 0.108 |
| Cross-lagged paths | | | | | | | | | | | | |
| SMSCF T1 → LS T2 | -0.131 | 0.110 | 0.236 | -0.140 | 0.110 | 0.204 | -0.135 | 0.084 | 0.108 | -0.131 | 0.082 | 0.108 |
| SMSCF T2 → LS T3 | -0.148 | 0.219 | 0.500 | -0.113 | 0.167 | 0.500 | -0.135 | 0.084 | 0.108 | -0.131 | 0.082 | 0.108 |
| LS T1 → SMSCF T2 | -0.215 | 0.119 | 0.070 | -0.212 | 0.118 | 0.072 | -0.228 | 0.093 | **0.015**\* | -0.240 | 0.091 | **0.008**\*\* |
| LS T2 → SMSCF T3 | -0.254 | 0.162 | 0.118 | -0.275 | 0.130 | **0.034**\* | -0.228 | 0.093 | **0.015**\* | -0.240 | 0.091 | **0.008**\*\* |
| Covariance | | | | | | | | | | | | |
| Between-person | -0.114 | 0.064 | 0.073 | -0.114 | 0.065 | 0.079 | -0.112 | 0.063 | 0.077 | -0.112 | 0.062 | 0.070 |
| Intra-wave correlation T1 | -0.012 | 0.048 | 0.802 | -0.012 | 0.049 | 0.803 | -0.018 | 0.036 | 0.628 | -0.016 | 0.035 | 0.650 |
| Intra-wave correlation T2 | -0.142 | 0.048 | **0.003**\*\* | -0.143 | 0.048 | **0.003**\*\* | -0.141 | 0.048 | **0.003**\*\* | -0.144 | 0.049 | **0.003**\*\* |
| Intra-wave correlation T3 | -0.023 | 0.038 | 0.549 | -0.021 | 0.036 | 0.557 | -0.018 | 0.032 | 0.566 | -0.015 | 0.031 | 0.613 |

$N_{T1}$ = 594, $N_{T2}$ = 410, $N_{T3}$ = 329, SMSCF = Social media self-control failure, LS = Life satisfaction.

\*\*\**p* < .001,

\*\**p* < .01,

\**p* < .05

significant only at T2 and in the model which constrained the autoregressive path to be equal, indicating that individuals scored higher than his/her expected scored on SMSCF scored lower than his/her expected score on subjective vitality.

At the between-person level, SMSCF and subjective vitality were significantly correlated in all constraint conditions, meaning that regardless of within-person change, the traits-like differences between individuals on SMSCF and subjective vitality were negatively correlated.

Adding age, gender, frequency of visit on social media and time spent on social media as control variables did not affect any of the results.

### Reciprocal model for social media self-control failure and life satisfaction

The models under all constraint conditions showed good fit (see Table 5). At the within-person level, a negative and significant cross-lagged path was found from life satisfaction in T2 to SMSCF in T3 in three constraint conditions, suggesting that more life satisfaction in T2 predicted less SMSCF in T3. Also, life satisfaction in T1 predicted SMSCF in T2 in the models that constrained the cross-lagged paths to be equal only or constrained both autoregressive and cross-lagged paths to be equal.

There was a significant and negative intra-wave residual correlation between SMSCF at T2 and life satisfaction at T2 in all constraint conditions, showing that a higher than expected score on SMSCF was related to a lower than expected score on life satisfaction. At the between-person level, there was no significant correlation between SMSCF and life satisfaction in all constraint conditions.

Controlling for age, gender, frequency of visit on social media, and time spent on social media did not affect the results regarding the associations between the trait-like individual differences.

The results regarding within-person change remained the same after controlling for age and frequency of visit on social media. However, after controlling for gender, in the model which constrained the autoregressive paths to be equal, there was no longer a significant path from life satisfaction T2 to social media self-control failure T3. Also, controlling for time spent on social media rendered the paths from life satisfaction T1 to social media self-control failure T2, and from T2 to T3 were no longer significant in the model that constrained the cross-lagged paths.

## Exploratory analysis: Mindfulness and wellbeing

We also explored the relationships between mindfulness and wellbeing, again using different constraints (or no constraints). The models for both subjective vitality and life satisfaction showed acceptable fit (see Table 9).

No reciprocal effects were found between mindfulness and the two wellbeing indicators in all constrained models, yet several cross-lagged paths were significant. Specifically, mindfulness predicted subjective vitality in the model that constrained both autoregressive paths and cross-lagged paths to be equal. Conversely, subjective vitality predicted mindfulness in the model that constrained the cross-lagged paths to be equal. Moreover, mindfulness predicted

**Table 9. Model comparisons for different constraint conditions in the exploratory analysis.**

| Model | | $\chi^2$ | $df$ | $p$ | CFI | TLI | SRMR | RMSEA [90% CI] | Model comparison | $\Delta\chi^2$ | $\Delta df$ | $\Delta p$ |
|---|---|---|---|---|---|---|---|---|---|---|---|---|
| Mindfulness-Subjective wellbeing | A1 | 0.540 | 1 | 0.462 | 1.000 | 1.004 | 0.005 | 0.000 [0.000, 0.097] | | | | |
| | B1 | 16.040 | 3 | 0.001 | 0.992 | 0.960 | 0.031 | 0.086 [0.048, 0.129] | A1 vs. B1 | 15.5 | 2 | < 0.001 |
| | C1 | 0.078 | 3 | 0.994 | 1.000 | 1.010 | 0.002 | 0.000 [0.000, 0.000] | A1 vs. C1 | 11.54 | 2 | 0.003 |
| | D1 | 18.082 | 5 | 0.003 | 0.992 | 0.976 | 0.029 | 0.066 [0.035, 0.101] | A1 vs. D1 | 17.54 | 4 | 0.002 |
| Mindfulness-Life satisfaction | A2 | 0.112 | 1 | 0.738 | 1.000 | 1.008 | 0.003 | 0.000 [0.000, 0.076] | | | | |
| | B2 | 0.337 | 3 | 0.953 | 1.000 | 1.008 | 0.005 | 0.000 [0.000, 0.000] | A2 vs. B2 | 0.23 | 2 | 0.894 |
| | C2 | 1.685 | 3 | 0.640 | 1.000 | 1.004 | 0.008 | 0.000 [0.000, 0.055] | A2 vs. C2 | 1.57 | 2 | 0.455 |
| | D2 | 2.721 | 5 | 0.743 | 1.000 | 1.004 | 0.014 | 0.000 [0.000, 0.040] | A2 vs. D2 | 2.61 | 4 | 0.625 |

SMSCF = Social media self-control failure, Model A: unconstrained model, Model B: constrained autoregressive paths to be equal, Model C: constrained cross-lagged paths to be equal, Model D: constrained both autoregressive paths and cross-lagged paths to be equal.

*$p < .05$

**Table 10. Reciprocal relationships between mindfulness and subjective vitality.**

| SMSCF—Vitality | Unconstrained model (df = 1) | | | Constrained autoregressive paths (df = 3) | | | Constrained for cross-lagged paths (df = 3) | | | Constrained autoregressive and cross-lagged paths (df = 5) | | |
|---|---|---|---|---|---|---|---|---|---|---|---|---|
| | b | SE | p | b | SE | p | b | SE | p | b | SE | p |
| Autoregressive paths | | | | | | | | | | | | |
| MF T1 → MF T2 | 0.247 | 0.195 | 0.206 | 0.360 | 0.203 | 0.076 | -0.122 | 0.099 | 0.217 | 0.396 | 0.157 | **0.012*** |
| MF T2 → MF T3 | 0.435 | 0.201 | **0.030*** | 0.360 | 0.203 | 0.076 | -0.060 | 0.147 | 0.683 | 0.396 | 0.157 | **0.012*** |
| SV T1 → SV T2 | 0.102 | 0.108 | 0.344 | -0.091 | 0.174 | 0.600 | 0.224 | 0.167 | 0.181 | -0.042 | 0.125 | 0.735 |
| SV T2 → SV T3 | -0.807 | 0.652 | 0.216 | -0.091 | 0.174 | 0.600 | 0.164 | 0.175 | 0.348 | -0.042 | 0.125 | 0.735 |
| Cross-lagged paths | | | | | | | | | | | | |
| MF T1 → SV T2 | 0.168 | 0.137 | 0.221 | 0.183 | 0.293 | 0.532 | -0.135 | 0.084 | 0.108 | 0.307 | 0.096 | **0.001**** |
| MF T2 → SV T3 | 0.935 | 0.524 | 0.074 | 0.408 | 0.220 | 0.064 | -0.135 | 0.084 | 0.108 | 0.307 | 0.096 | **0.001**** |
| SV T1 → MF T2 | 0.212 | 0.115 | 0.066 | 0.091 | 0.134 | 0.498 | -0.228 | 0.093 | **0.015*** | 0.108 | 0.098 | 0.274 |
| SV T2 → MF T3 | -0.012 | 0.244 | 0.959 | 0.113 | 0.166 | 0.497 | -0.228 | 0.093 | **0.015*** | 0.108 | 0.098 | 0.274 |
| Covariance | | | | | | | | | | | | |
| Between-person | 0.176 | 0.050 | **0.000***** | 0.191 | 0.057 | **0.001**** | -0.112 | 0.063 | 0.077 | 0.175 | 0.048 | **0.000***** |
| Intra-wave correlation T1 | 0.039 | 0.045 | 0.389 | 0.028 | 0.057 | 0.626 | -0.018 | 0.036 | 0.628 | 0.049 | 0.044 | 0.260 |
| Intra-wave correlation T2 | 0.090 | 0.025 | **0.000***** | 0.070 | 0.029 | **0.014*** | -0.141 | 0.048 | **0.003**** | 0.076 | 0.023 | **0.001**** |
| Intra-wave correlation T3 | 0.041 | 0.033 | 0.213 | 0.053 | 0.024 | **0.026*** | -0.018 | 0.032 | 0.566 | 0.041 | 0.018 | **0.021*** |

$N_{T1}$ = 594, $N_{T2}$ = 410, $N_{T3}$ = 329, Mindful = Mindfulness, SV = Subjective vitality.

***$p < .001$,

**$p < .01$,

*$p < .05$

life satisfaction in the model that constrained either the cross-lagged paths or both the autoregressive path and the cross-lagged paths to be equal.

At the between-person level, there was a positive correlation between mindfulness and subjective vitality in all constraint conditions except for the model that only constrained the cross-lagged paths to be equal. There was also a positive correlation between mindfulness and life satisfaction in all models (see Tables 10 and 11).

## Discussion

Living in a 'permanently online' world [1] generates self-control problems for many social media users. Frequent failures to control one's social media use may result in lower mindfulness and lower wellbeing, but may also result from lower mindfulness and lower wellbeing. Using a 3-wave longitudinal design, the present study examined the reciprocal relationships between social media self-control failure, mindfulness and wellbeing.

We expected that social media self-control failure and mindfulness would reciprocally influence each other over time. This hypothesis was only partially confirmed. We found that people's social media self-control failure at T1 resulted in less mindfulness at T2, which led to subsequent social media self-control failure in T3. However, this effect was only found for two out of the four constraint conditions, in the models in which the T1-T2 and T2-T3 cross-lagged effects were constrained to be equal, no cross-lagged effects were found. It should be noted that the partial reciprocal relation neither fully supports nor rejects our hypothesis H1. We did not preregister how we would apply constraints to our model, and how we would treat differences between the results under each condition. Neither did we set clear rules for whether a reciprocal effect between two (instead of three) waves supports a reciprocal effect. Since the

**Table 11. Reciprocal relationships between mindfulness and life satisfaction.**

| | Unconstrained model (df = 1) | | | Constrained autoregressive paths (df = 3) | | | Constrained cross-lagged paths (df = 3) | | | Constrained autoregressive and cross-lagged paths (df = 5) | | |
|---|---|---|---|---|---|---|---|---|---|---|---|---|
| | *b* | *SE* | *p* | *b* | *SE* | *p* | *b* | *SE* | *p* | *b* | *SE* | *p* |
| Autoregressive paths | | | | | | | | | | | | |
| MF T1 → MF T2 | 0.353 | 0.226 | 0.119 | 0.369 | 0.149 | **0.013*** | 0.350 | 0.205 | 0.088 | 0.401 | 0.139 | **0.004**** |
| MF T2 → MF T3 | 0.370 | 0.161 | **0.022*** | 0.369 | 0.149 | **0.013*** | 0.418 | 0.122 | **0.001**** | 0.401 | 0.139 | **0.004**** |
| LS T1 → LS T2 | 0.129 | 0.186 | 0.488 | 0.109 | 0.157 | 0.488 | 0.199 | 0.156 | 0.204 | 0.139 | 0.141 | 0.324 |
| LS T2 → LS T3 | 0.036 | 0.308 | 0.906 | 0.109 | 0.157 | 0.488 | 0.085 | 0.223 | 0.702 | 0.139 | 0.141 | 0.324 |
| Cross-lagged paths | | | | | | | | | | | | |
| MF T1 → LS T2 | 0.326 | 0.254 | 0.200 | 0.335 | 0.201 | 0.096 | 0.359 | 0.181 | **0.047*** | 0.354 | 0.167 | **0.034*** |
| MF T2 → LS T3 | 0.288 | 0.335 | 0.390 | 0.223 | 0.212 | 0.295 | 0.359 | 0.181 | **0.047*** | 0.354 | 0.167 | **0.034*** |
| LS T1 → MF T2 | 0.018 | 0.122 | 0.886 | 0.004 | 0.099 | 0.965 | 0.100 | 0.080 | 0.209 | 0.111 | 0.075 | 0.140 |
| LS T2 → MF T3 | 0.181 | 0.141 | 0.199 | 0.185 | 0.104 | 0.074 | 0.100 | 0.080 | 0.209 | 0.111 | 0.075 | 0.140 |
| Covariance | | | | | | | | | | | | |
| Between-person | 0.213 | 0.074 | **0.004**** | 0.215 | 0.069 | **0.002**** | 0.193 | 0.078 | **0.013*** | 0.191 | 0.076 | **0.011*** |
| Intra-wave correlation T1 | 0.050 | 0.066 | 0.450 | 0.050 | 0.066 | 0.449 | 0.077 | 0.069 | 0.262 | 0.077 | 0.069 | 0.262 |
| Intra-wave correlation T2 | 0.116 | 0.045 | **0.010*** | 0.112 | 0.039 | **0.004**** | 0.134 | 0.040 | **0.001**** | 0.137 | 0.039 | **0.000***** |
| Intra-wave correlation T3 | 0.080 | 0.033 | **0.015*** | 0.078 | 0.028 | **0.005**** | 0.067 | 0.025 | **0.007**** | 0.068 | 0.023 | **0.003**** |

$N_{T1}$ = 594, $N_{T2}$ = 410, $N_{T3}$ = 329, MF = Mindfulness, LS = Life satisfaction.

***$p < .001$,

**$p < .01$,

*$p < .05$

falsification criteria for our hypotheses were not fully detailed beforehand, the results should be interpreted with care. This also suggests that future research using RI-CLPM should include the criteria for applying constraints and the falsification criteria while designing the study. We will come back to the effects of applying constraints to the model later on in this discussion.

As was expected, the results also revealed a between-person association of social media self-control failure and mindfulness, showing that people who failed more often in controlling their social media use also tended to show lower mindfulness. In the RI-CLPM, this is indicated by a correlation between the stable, trait-like components of social media self-control failure and mindfulness. This result is consistent with recent findings showing that trait self-control was associated with higher mindfulness [27], as well as existing evidence that being permanently engaged in online interaction was related to lower mindfulness (e.g., [16]), but also adds to these findings. Applying the RI-CLPM approach, our findings showed not only that between persons, mindfulness and social media self-control failure covary, but also that within persons, changes in social media self-control failure predict changes in mindfulness and vice versa, above and beyond the between-person associations.

Yet, the results did not provide evidence for a fully reciprocal relation between social media self-control failure and mindfulness. One possible explanation of the partially confirmed reciprocal relationship might be that the reciprocal model could be asymmetric, such that a certain path progresses concurrently rather than successively [56]. Hence, a 4-month interval might be too long to observe the interplay of mindfulness on social media self-control failure. This perspective is supported by a previous study comparing RI-CLPM models with different time intervals. Specifically, a reciprocal effect was found between media multitasking and attention problems only for the model with a 3-month interval, but not for the model with a 6-month interval [18]. Even for examining associations between personality traits, it is suggested that a

reciprocal model with a smaller time lag is more appropriate [57]. We set the interval between each survey at four months based on the study by Suh, Diener, and Fujita (1996), which suggests that one should have sufficient time between the measurement times in order to detect potential fluctuations in life satisfaction. Suh et al. used 3 months for their time lag [58], we chose four months between the measurements in order to avoid the time 2 survey to be in the holiday season. Overall, we only detected part of a possible reciprocal effect in the relationship between social media self-control failure and mindfulness during an eight-month period. Nevertheless, Slater et al. (2007) further proposed that given ongoing continuous mutual influence, the reciprocal effects might be prospectively observed [56]. In this case, adding more waves might increase the probability to observe the expected reciprocal effects. Therefore, future research could vary the time lag and time length of the reciprocal models.

Examining the reciprocal effects may contribute to enrich the empirical evidence for the reinforced spiral model of media effects [56, 59]. The reinforced spiral model of media effects proposes that selected exposure to certain kinds of media content could affect people's attitude or cognition, which will increase the possibility of future exposure to the same kinds of media, and vice versa [59]. Baumgartner et al. (2017) extended the boundary of this model by examining the reciprocal relationship between specific types of media use (i.e., media multitasking) and individual difference factors (i.e., attention problems) [18]. The present study further studied a problematic form of social media use and its reciprocal relationships with mindfulness and wellbeing. Though no full reciprocal paths were observed, our findings may provide insights into how these variables mutually develop over time, and their possible causal relationships.

The within-person relationship between social media self-control failure and mindfulness has implications for interventions to improve self-control in (social) media use. Elkins-Brown et al. (2017) propose that mindfulness may sensitize people to affective change when goal-conflict occurs, which benefits goal-directed behaviors and self-control. Indeed, in the clinical field, mindfulness training was demonstrated to improve healthy behaviors, such as healthy sleep habits, eating practice and physical exercise [60]. The results imply that mindfulness training may also be of help to strengthen media related self-control. Conversely, our findings suggest that improving social media related self-control might also improve mindfulness. For many users, social media distractions happen numerous times a day. Resisting social media temptations could be seen as a way to deploy attention awareness to frequently occurring situations which give rise to mindless social media behaviors, such as habitual social media checking, or social media induced mind-wandering. This practice could help to increase mindfulness in general.

Partly in line with our expectation, the findings showed a between-person relationship between social media self-control failure and subjective vitality. This indicates that those who perceived more failure in controlling their social media use also felt the loss of subjective vitality, such as a lack of energy and the tendency to be easily drained [53]. Similar evidence was shown in cross-sectional research regarding the negative association between social media self-control failure and subjective vitality [7]. This result is also consistent with evidence showing that media-related entertainment benefits subjective vitality only when people perceive sufficient control [61]. However, with regard to our research question "will social media self-control failure and wellbeing reciprocally and negatively influence each other?" the results show no significant reciprocal relationship between social media self-control failure and subjective vitality, indicating that social media self-control failure did not significantly affect subjective vitality over time, or vice versa, and that any claim about a causal relationship needs to be treated with caution.

The relationship between social media self-control failure and affective wellbeing may also depend on the type of gratifications expected to gain from social media use. For instance, Du et al. (2019) found that when social media use was expected to satisfy social gratifications, people were more successful in controlling their social media use [2]. However, when social media use was expected to satisfy information seeking and entertainment gratifications, people more easily failed to control their social media use. In this study, we did not distinguish between the type of social media use and its related self-control problems. Thus, it could be that self-control failures induced by specific kinds of social media use might not necessarily reduce wellbeing, which presents an interesting option to explore in future research.

Unexpectedly, the results with regard to life satisfaction show a different pattern: no significant between-person correlation was observed, yet three out of four models show that life satisfaction predicts subsequent social media self-control failure. No support for a reciprocal model was found, none of the cross-lagged paths from social media self-control failure to life satisfaction was significant (although most coefficients were close to reaching significance). The lack of support for a path from social media self-control failure to life satisfaction implies that social media self-control failure does not necessarily lead to a long-term decline in overall evaluation of one's life. This might be because being always engaged online provides rapid and timely support for people. As time goes by, their online social capital could be slowly accumulated and enhanced, as was demonstrated in a previous study that showed that a larger network on social media and a greater perceived audience online can promote a positive evaluation of one's life [62]. Moreover, since life satisfaction is a product of cognitive evaluation, some self-related cognitive factors might also play an important role in determining life satisfaction. For instance, Cummins and Nistico (2002) found that the evaluation of life satisfaction can be largely biased by self-esteem and optimism. Due to these factors, people may think or report that they are happier than they really are (the "life satisfaction homeostasis") [63]. Thus, to what extent these factors influence the link between social media self-control failure and life satisfaction at a between-level should be considered.

Vice versa, life satisfaction, but not subjective vitality predicted subsequent social media self-control failure. In half of the constraint conditions, low life satisfaction in T1 predicted social media self-control failure in T2, and in three out of four constraint conditions, the same predictive pattern was seen from T2 to T3. These findings indicate that having a negative evaluation of one's life in general might increase the potential for more social media self-control failure. This is in line with recent findings showing that increased life satisfaction predicted slightly lower social media use [48].

In addition, an exploratory analysis revealed a between-person association between mindfulness and both subjective vitality (except for the model that constrained the cross-lagged paths to be equal), and life satisfaction. No reciprocal relationship was found between mindfulness and subjective vitality, or mindfulness and life satisfaction. Nevertheless, the analyses revealed a significant cross-lagged path from mindfulness to subjective vitality, as well as a reversed cross-lagged path from subjective vitality to mindfulness in one out of four constraint models, respectively. Besides, mindfulness also predicts life satisfaction in two constraint models, suggesting that mindfulness predicts wellbeing more so than vice versa. Though no clear conclusion can be reached based on the data, the results imply an intermediate role of mindfulness between impaired social media self-control and decreased subjective vitality. Future research should explore how and to what extent mindfulness serves as a linkage between these two variables.

In additional (not preregistered) analyses, we controlled for age, gender and social media use in each of the main models. Results showed that the relationship of the trait-like individual differences between social media self-control failure and mindfulness remains almost

unaffected by including the control variables, yet within-person growth of these constructs was more affected. The same pattern can be seen in the relationship between social media self-control failure and life satisfaction. The results indicate that the robustness of the results is different at the within-person level and between-person level given the influence of control variables. This further stresses that when examining the longitudinal relationships between these psychological constructs, it is necessary to apply an approach to distinguish the trait-like and state-like variances.

## Strengths and limitations

The main strength of the present study is that it offered a first test of the reciprocal influences of social media self-control failure on mindfulness and wellbeing. Using a RI-CLPM, we assessed the stable between-person variance of the psychological constructs and their within-person change separately. This permits taking a deeper look into the relationships between social media self-control failure, mindfulness and wellbeing over time. Another strength of the study is that we focused on social media users' everyday self-control problems of social media use. We measured social media self-control failure that people may experience every day, rather than pathological forms of social media behaviors. Also, we included mindfulness as a key variable. Only a few studies have looked at the role of mindfulness in media use, none of which employed the RI-CLPM approach we employed on our study.

Yet, some limitations of the present study should be noted. First, although associations between social media self-control failure, mindfulness and subjective vitality were found using a longitudinal design, we did not experimentally manipulate these variables in order to observe their causal relationships. Thus, we cannot rule out other variables which might confound the associations between social media self-control failure, mindfulness, and wellbeing. More rigorous controlled methods could be used in future research to examine the causal relations between these variables. Second, all constructs in the study were assessed by self-report measurements, which could affect the validity in real-life scenarios. As was mentioned before, there is evidence that life satisfaction can be largely biased by one's positive cognitions pertaining to the self, such as optimism [63]. Similarly, self-reported social media self-control failure could be biased by memory, as for many people this failure occurs frequently and may be difficult to remember correctly. Thus, using a more in-situ approach (e.g., experience sampling [39]) would provide a more objective record of real-life social media self-control failure."

In addition, participants were instructed to indicate their current level of mindfulness and subjective vitality. For social media self-control failure and life satisfaction, we did not specify the time frame of the questions. Thus, some participants might have been prone to report their trait-like, general level of social media self-control failure and life satisfaction. Yet, after establishing the trait-like individual differences of these variables in the RI-CLPM, we still found the within-person effect of social media self-control failure on mindfulness, and the within-person association between life satisfaction and future social media self-control failure. Nevertheless, it could be that the instruction of the measurements reduced within-person variance in some of our measurements. Therefore, future research should take into account the time frame of the instruction regarding the measurements.

Importantly, we have chosen to test different models in which we varied the constraints on the different paths in the model. In our pre-registration, we did not specify how we would deal with constraints. While analyzing the data, we found different model specifications to lead to different outcomes. We have chosen to report all the results since there are arguments both pro and con applying constraints. The time lags between the measurements were the same between each wave (4 months), which is an argument for constraining the paths between T1,

T2 and T3 to be equal. Yet the questionnaires were sent out at different times of the year, which may undermine the stability assumed in constrained models. For example, the time between T2 and T3 included a summer holiday, which may affect habits in social media use and which may also affect wellbeing. Since models with and without constraints showed hardly any difference in model fit, we estimated models with or without constraints in order to prevent the risk that one specific analysis yields false positive results (cf. [48]). The different findings for different models also raise the awareness that applying constraints affects the estimation of the coefficients and the conclusions of RI-CLPM. Future research should be cautious about the flexibility in using this model, particularly when the purpose of applying such a model is to understand "the true underlying reciprocal process that takes place at the within-person level" [20].

Besides, the current research is limited by using mean score of each variable to fit the RI-CLPM, based on the rather strong assumption of no measurement error. However, this could bias lagged-parameter estimates, which weakens the explanatory power of the findings [55]. We also tried the latent variable model in analyzing the data. Following Hamaker & Mulder (2020), estimating the latent model is a 4-step process which helps build measurement invariance. However, results showed estimation problems during each step, which made us decide to hold on to using mean scores as indicators in our models. Future research could use multiple indicators of each variable to fit the RI-CLPM, or use alternative reciprocal models that account for measurement error, such as the stable trait autoregressive trait and state model (see Usami et al. (2019) for a review of different types of reciprocal models) [64].

Taking these limitations into account, we are cautious about overstating our conclusions. We found evidence for between-person associations between the time-invariant, trait like components of social media self-control failure on the one hand, and mindfulness and subjective vitality (but not life satisfaction) on the other hand, across different types of models. Based on the intra-wave correlations between the residuals, we also find consistent (across models, across the three different measurements) evidence that shows that within-person changes in social media self-control failure are accompanied by changes in mindfulness. The evidence for a reciprocal effect between social media self-control failure and mindfulness was contingent both on whether and which constraints were applied and on time interval (T1-T2, or T2-T3). The effect over time of life satisfaction on social media self-control failure was consistent across the two-time intervals, but could not be established in a model without constraints.

This leads us to conclude that although our findings show consistent evidence for between-person associations between social media self-control failure, mindfulness and wellbeing, and how they change within persons, at this stage support for any causal relation is preliminary and needs further research.

## Conclusion

The present study focused on social media self-control failure which many people experience on a daily basis. Using a 3-wave longitudinal design, we looked at how social media self-control failure is related to mindfulness and wellbeing. The results indicated that between persons, failure in controlling one's social media use was stably associated with a lower mindfulness and lower subjective vitality during an eight-month period. The reciprocal effects between social media self-control failure and mindfulness were partially observed, indicating that social media self-control failure preceded lower mindfulness after four months, which in turn preceded social media self-control failure four months later. Although at this stage it is too early to come to firm evidence-based recommendations regarding practical implications, in time our study may help to inform interventions to prevent social media self-control failure. Given

the indications for a reciprocal relation between mindfulness and social media self-control failure, our findings imply that interventions may be aimed both at increasing mindfulness and/or directly at strengthening social media self-control.

## Note

a. The original hypothesis in the pre-registration was "Social media self-control failure will negatively predict mindfulness, subjective vitality and life satisfaction cross-sectionally."

b. The other variables collected in T1 were: immediate gratifications from social media use, habitual checking of social media, perceived ubiquity of social media, and notifications received from social media.

c. We also identified 5 "speeders" who completed the survey faster than 40% of the median completion time [65]. In total, 5 participants completed the survey faster than 156.4 seconds in time 1 survey, or 84.4 seconds in time 2 survey, or 78.8 seconds in time 3 survey. We rechecked the results after excluded these participants, no differences were found regarding the main results of the study.

## Supporting information

**S1 File. Confirmatory analysis of the scales in the reciprocal model.**
(TXT)

**S2 File. Results including control variables.**
(DOCX)

## Author Contributions

**Conceptualization:** Jie Du, Peter Kerkhof, Guido M. van Koningsbruggen.

**Data curation:** Jie Du.

**Formal analysis:** Jie Du.

**Investigation:** Jie Du.

**Methodology:** Jie Du, Peter Kerkhof, Guido M. van Koningsbruggen.

**Supervision:** Peter Kerkhof.

**Validation:** Peter Kerkhof.

**Writing – original draft:** Jie Du.

**Writing – review & editing:** Peter Kerkhof, Guido M. van Koningsbruggen.

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
