## [Decision Letter · Decision Letter 0]

22 Oct 2020

PONE-D-20-21281

The reciprocal relationships between social media self-control failure, mindfulness and wellbeing: A longitudinal study

PLOS ONE

Dear Dr. Du,

Thank you for submitting your manuscript to PLOS ONE. After careful consideration, we feel that it has merit but does not fully meet PLOS ONE’s publication criteria as it currently stands. Therefore, we invite you to submit a revised version of the manuscript that addresses the points raised during the review process.

I have now received two reviews of your manuscript from experts in the field. Both reviewers felt the research was valuable and addressed an important gap in the current literature on social media, self-control and well-being, especially with respect to its design and the Open Science approach that was taken in conducting the research.

Although both reviewers noted that they enjoyed reading your paper, they also noted several areas that needed further attention and invested considerable time in providing extremely detailed and constructive comments, along with suggested references, aimed at improving the paper. For example, both reviewers suggest that the paper would benefit from greater elaboration of the theoretical rationale and mechanisms that underpin the proposed relationships among the key variables. Similarly, Reviewer 1 raises some important issues with respect to how your constructs were operationalised that have important implications for the assumption underlying the statistical modelling that as used. These are just some of the important issues that deserve close attention, and I will not repeat all of the reviewer’s recommendations for improvement here. I do believe that their concerns are substantial enough that the paper cannot be accepted in its present form.

However, it may be possible that with careful attention to their comments, a substantially revised paper could be considered for publication. Given this, I would like to invite you to make substantial revisions in line with the reviewers’ comments and resubmit a revised version of the manuscript. Please include a cover letter detailing how you have dealt with each of the comments.

Thank you for considering PLOS ONE as an outlet for your work and I look forward to receiving your revised manuscript.

We look forward to receiving your revised manuscript.

Kind regards,

Fuschia M. Sirois, PhD

Academic Editor

PLOS ONE

Journal Requirements:

Reviewers' comments:

Reviewer's Responses to Questions

**Comments to the Author**

1. Is the manuscript technically sound, and do the data support the conclusions?

Reviewer #1: Yes

Reviewer #2: Yes

2. Has the statistical analysis been performed appropriately and rigorously? 

Reviewer #1: Yes

Reviewer #2: Yes

3. Have the authors made all data underlying the findings in their manuscript fully available?

Reviewer #1: No

Reviewer #2: No

4. Is the manuscript presented in an intelligible fashion and written in standard English?

Reviewer #1: Yes

Reviewer #2: Yes

5. Review Comments to the Author

Reviewer #1: I applaud the authors for a great paper, which I enjoyed reading, and a methodologically sound study. I have only a few recommendations to improve your work. The most important one is that you should delve deeper into the theoretical connection of the constructs of interest (beyond only quoting empirical findings), thereby enriching your theoretical background and discussion/outlook.

(1) Theoretical rationale

a) Social media self-control failure (SMSCF) is an interesting construct. Nevertheless, we still know very little about it. First, I think you should make a stronger case for its relevance. Why is it important to investigate SMSCF? Why is it superior to other constructs (e.g., trait self-control, procrastination)? etc.

b) Second, the dynamic and reciprocal relationships are at the core of your assumptions and modeling. Theoretically, this part definitively needs more elaboration. Your assumptions comprise important deviations from Slater’s (2007, 2015). For instance, in Slater’s (2007) model, the central variables are media use as well as cognitive and behavioral variables (e.g., attitudes, behavior, identity). Your model focuses on SMSCF (which already is a cognitive effect of media use), mindfulness, and well-being. As a minimum, I would have expected some words on how Slater’s reinforcing spirals transfers to your model.

c) From a theoretical perspective, I’d also like to read more about the theoretical connections between SMSCF, mindfulness, and well-being. Although you provide empirical evidence, little is said in terms of why they should be related (e.g., how can SMSCF theoretically affect trait mindfulness).

d) From my perspective, psychological well-being is misconceived. What you refer to is often understood as subjective or hedonic well-being, whereas psychological or eudaimonic well-being often relates to other theoretical concepts and operationalizations (Keyes et al., 2002; Martela & Sheldon, 2019; Ryff & Singer, 1998). Thus, the corresponding parts need clarification.

e) I’m not fully convinced of your elaboration on stability and change. I think you need to flesh out the trait and state components of the constructs you’ve theorized and also operationalized. For instance, all the measures you’ve applied can be considered as capturing state and trait components. Thus, the random intercept cross-lagged panel model (RI-CLPM) is an appropriate way to account for this. But I’m not sure if the operationalizations are those of variables that are really susceptible to change. All of them seem to rather tap into trait-like or generalizable statements than assess a specific state. There is no particular time reference in your measures—neither in the item stems nor in the response options. An item such as “How often do you give in to a desire to use social media even though your social media use at that particular moment makes you delay other things you want or need to do?” with response options ranging from “never” to “always” doesn’t make clear if this measures a situational behavior or a more stable personality trait. As you’ve shown (Du et al., 2018), the scale has high test–retest reliability and thus seems to rather measure a trait variable. Similarly, this also holds true for the mindfulness and the life satisfaction measure. Subjective vitality could be more prone to change but this may totally depend on your instruction (e.g., rate how you feel today, the last week, the time since the last questionnaire etc.). Still, there could be a large portion of trait variance influencing the responses.

In short, the role of trait and state components is appropriately accounted for in the RI-CLPM but neither discussed theoretically nor operationalized in a suitable way.

(2) Methodological soundness

Besides 1e), there’s little to moan with regard to methods, although it would have been great to inspect the analyses’ code and check for reproducability (see also 4b).

a) A bit more methodologically sophisticated approaches would check the construct validity of the constructs using a confirmatory factor analysis and also test for measurement invariance across measurement occasions (Brown, 2015; Geiser et al., 2015). Perhaps not essential, but could be used to show distinctness of the constructs within and across waves and comparable meaning of the measures across the three waves.

b) I’m not sure why Facebook use was measured. Moreover, in the study using the t1 data (Du et al., 2019), you reported the same items including the terms “social media” not Facebook.

c) In general, I wondered why no control variables were included, at least in the unpreregistered exploratory analyses. Your own studies showed correlations with social media use, age, and gender. Wouldn’t it be useful to check their relation to the constructs of interest as well?

(3) Discussion

a) I’d like to see a more thorough discussion against the backdrop of recent findings (e.g., Liu et al., 2019; Orben et al., 2019; Stavrova & Denissen, 2020). Orben et al. (2019) is already briefly mentioned in some parts of the discussion. You could flesh out these sections.

b) Moreover, the “social media” is a very broad term. Thus, self-control failures could refer to many different aspects of using social media features. On the one hand, given the results of Du et al. (2019), it could be interesting to discuss which features are theoretically positively related to well-being (e.g., social gratifications to satisfy belongingness or information gratifications to reduce uncertainty) but less to self-control failure, and vice versa. On the other hand, a more nuanced discussion on the potentially causal relationship between well-being and self-control failure would also help to gain insight into the relevant facets of well-being. The very broad and general Satisfaction with Life Scale (SWLS) is not necessarily directly linked to SMSCF. However, as SWLS is a cognitive indicator of subjective well-being, it could be that self-concept relevant aspects are mediating this relationship. This is just a fictitious example. As the theoretical background could be improved, so could be the discussion. More speculation about the mechanisms and conditions of the connections between SMSCF, mindfulness, life satisfaction, and vitality are necessary.

c) Your limitation section starts fast-forwarding to the model constraints. Why are experimental longitudinal designs are badly needed? Why is it important to pay attention to actual and not only self-reported media use? Why is it useful to combine media use and experienced loss of self-control in a single measure? etc. These questions deserve more attention.

(4) Open Science

Besides preregistering the present study, the present paper does not address all of proposed Open Science practices (e.g., Dienlin et al., 2020).

a. Preregistering: You’ve pregistered your study at aspredicted.org. This is great. While reading the preregistration, however, I thought that it could have been beneficial to split it in two preregistrations as it has clearly been your plan from the beginning to test and publish the findings concerning your first research goal (“predictors of social media self-control failure”) in a separate paper (Du et al., 2019). Please cite this paper also on p. 8, where you refer to this publication. Moreover, I wonder whether some of those constructs you’ve examined (and found to be associated with SMSCF!) in that paper could be valuable control variables at t1 (e.g., age, online vigilance) for your present analyses. Finally, the wording of the hypotheses is not exactly the same. In the present paper, please explain why you deviate from the preregistration.

b. Open data & material: I encourage you to upload your data and material as online supplementary material, so that both readers and reviewers can rerun analyses or check the operationalizations and implementation of the study in more detail. If it should not be possible to upload the data (e.g., due to data protection laws), the material (e.g., due to copyright infringements), and the analysis scripts, please state those explicit reasons why it is not possible. Until now, the OSF repository only includes the preregistration, the figure, and the tables—all of them are already included in your full paper submission.

c. Please list all the items that were used (either in an appendix or as online supplementary material; for example, on osf.io). In the OSF project of Du et al. (2019), you provide a well-organized codebook of all the relevant variables. This is clearly missing for this submission.

Minor issues:

– On p. 4, lines 56–58, you provide an example for a reinforced downward spiral (Orben et al., 2019). I’m not sure if this example suits your rationale because they found rather small effects and results were contingent on gender (a variable you do not include in your considerations). Moreover, the following sentence (lines 58–60) does not tell the reader why this would “provide a more complete picture.”

– p. 6: The study mentioned in Lines 113–115 (Bauer et al., 2017) is a longitudinal diary study, not a cross-sectional one.

 

References

Bauer, A. A., Loy, L. S., Masur, P. K., & Schneider, F. M. (2017). Mindful instant messaging. Mindfulness and autonomous motivation as predictors of well-being and stress in smartphone communication. Journal of Media Psychology, 29(3), 159-165. https://doi.org/10.1027/1864-1105/a000225

Brown, T. A. (2015). Confirmatory factor analysis for applied research (2nd ed.). Methodology in the social sciences. Guilford Press.

Dienlin, T., Johannes, N., Bowman, N. D., Masur, P. K., Engesser, S., Kümpel, A. S., Lukito, J., Bier, L. M., Zhang, R., Johnson, B. K., Huskey, R., Schneider, F. M., Breuer, J., Parry, D. A., Vermeulen, I., Fisher, J. T., Banks, J., Weber, R., Ellis, D. A., . . . Vreese, C. de (2020). An agenda for open science in communication. Journal of Communication, Article jqz052, 1–26. https://doi.org/10.1093/joc/jqz052

Du, J., Kerkhof, P., & van Koningsbruggen, G. M. (2019). Predictors of social media self-control failure: Immediate gratifications, habitual checking, ubiquity, and notifications. Cyberpsychology, Behavior and Social Networking, 22(7), 477–485. https://doi.org/10.1089/cyber.2018.0730

Du, J., van Koningsbruggen, G. M., & Kerkhof, P. (2018). A brief measure of social media self-control failure. Computers in Human Behavior, 84, 68–75. https://doi.org/10.1016/j.chb.2018.02.002

Geiser, C., Keller, B., Lockhart, G., Eid, M., Cole, D., & Koch, T. (2015). Distinguishing state variability from trait change in longitudinal data: The role of measurement (non)invariance in latent state-trait analyses. Behavior Research Methods, 47(1), 172–203. https://doi.org/10.3758/s13428-014-0457-z

Liu, D., Baumeister, R. F., Yang, C., & Hu, B. (2019). Digital communication media use and psychological well-being: A meta-analysis. Journal of Computer-Mediated Communication, 24(5), 259–273. https://doi.org/10.1093/jcmc/zmz013

Orben, A., Dienlin, T., & Przybylski, A. K. (2019). Social media's enduring effect on adolescent life satisfaction. Proceedings of the National Academy of Sciences of the United States of America, 116(21), 10226–10228. https://doi.org/10.1073/pnas.1902058116

Slater, M. D. (2007). Reinforcing spirals: The mutual influence of media selectivity and media effects and their impact on individual behavior and social identity. Communication Theory, 17(3), 281–303.

Slater, M. D. (2015). Reinforcing spirals model: Conceptualizing the relationship between media content exposure and the development and maintenance of attitudes. Media Psychology, 18(3), 370–395. https://doi.org/10.1080/15213269.2014.897236

Stavrova, O., & Denissen, J. (2020). Does using social media jeopardize well-being? The importance of separating within- from between-person effects. Social Psychological and Personality Science. Advance online publication. https://doi.org/10.1177/1948550620944304

Reviewer #2: Dear authors,

I read your manuscript with great joy. As you know, in the literature on social media, self-control, and well-being there is a profound lack of longitudinal studies that differentiate between- and within-person effects. Also, much of the literature is opaque on data, not up-to-date with regard to Open Science practices, and often suffers from claims that simply aren’t backed by data. You make an important contribution to that literature by addressing all of these gaps. You a) preregistered your research, b) separate within- and between-person effects, c) intend to share your data and materials, and d) are modest in your conclusions. Also, you manuscript was clearly written and a pleasure to read. My compliments on your work.

During my review, I carefully examined:

• All parts of the paper

• The preregistration on the OSF

I did not:

• Review the supplementary materials carefully (lack of time)

• Review the analysis (because you didn’t provide data or code)

I will list my thoughts, concerns, and suggestions for improvement chronologically.

Theory Section

First, the theoretical rationale wasn’t always clear to me. You define social media self-control failure and mindfulness, but it’s not clear what psychological mechanism links those two. Self-control in Psychology more and more takes the form of a motivational conflict that states that people constantly survey the value of options and contrast them with the value of their current activity/goal (Berkman et al., 2017; Kurzban et al., 2013). It’s not clear how social media self-control failure fits here. By definition, it’s a self-control process, but the mechanism behind it doesn’t become clear (yet) from the paper. You say that social media present a temptation and also speak of goal-conflict. But it’s not clear a) where that value comes from, b) in what circumstances value conflicts with another goal, c) under what circumstances people give in to that temptation. You state that this presents a “potential impairment of a psychological function that underlies intended and sustained attention”. But before you can turn to attention, it’s important to inform the reader what role attention plays in the self-control process. For example, we know that high-value options attract attention (e.g., Anderson et al., 2011). As far as I know, there’s no consensus on the role of attention within self-regulation (Hofmann & Van Dillen, 2012; Inzlicht & Berkman, 2015). It appears you use attention as an outcome of the self-regulation process: There’s a high-value option available that conflicts with my goal, so I turn my attention there. But I’m missing the explicit theoretical link here.

Also, you use attention as the link between self-control and mindfulness. In fact, you state: “Thus, mindfulness is a crucial psychological function for self-control in everyday life”. For one, this is a strong statement and I’d like to hear how mindfulness is such a central part of self-control. Conceptually, mindfulness isn’t just about self-control. As you’re aware, mindfulness describes more than just the awareness/attention component, but also evaluation etc. I’d invite the authors to elaborate here on which models of self-regulation and mindfulness they refer to. It’s unclear to me how mindfulness fits a self-regulation process other than potentially sharing a common, latent trait of attentional focus. If that’s not clear to readers, it’s not clear why you measured all facets of mindfulness and not just the attentional component.

Second, most of your discussion of empirical evidence refers to concepts, not the theoretical mechanism. For example, page 6, line 102: social media use predicted burnout for those low in mindfulness. This study shares variables with yours, but I’m not sure how it supports the mechanism you propose. Mindfulness in the study is a moderator, but you use that study as support for your claim that social media self-control failure affects mindfulness directly. Another example, same page, line 108: you cite evidence that multitasking and attention were reciprocally related. I don’t see how this informs your study, unless you explain how multitasking stands in relation to social media self-control failure and how attention stands in relation to mindfulness (as a whole concept, not just the attention part). I kindly ask the authors to elaborate on their theoretical model and show how previous evidence relates to that model more clearly. Right now, most of the studies you cite just support that the concepts you study are related, but not how or why.

Third, I ask you to consider removing all mentions of addiction from the manuscript or at least acknowledge that the concept is highly problematic and we should see it with skepticism. The literature is pretty clear that social media/internet/smartphone addiction simply is a mess of a concept/field (Aagaard, 2020; Abendroth et al., 2020; Panova & Carbonell, 2018; Satchell et al., 2020). It isn’t a clinically accepted diagnosis, so I strongly believe we should stop using it altogether until we have better evidence. I know that this is highly personal suggestion, so feel free to argue your stance in the revision letter. But I’d at least like to see an acknowledgment in the manuscript that studies use addiction, which is not a thing, but you consider it as self-control failure for reasons X and Y.

Fourth, your introduction is overall rather negative. I think it would make the paper more balanced and present a better overview of the literature for readers if you also outline the positive sides of social media (Antheunis et al., 2015; Bayer et al., 2015; Domahidi, 2018; Meier et al., in press). I think that would fit a discussion of the high value of social media because benefits and enjoyment from social media can potentially explain the high value that’s responsible for self-control failure.

Method & Results

First, I was disappointed to see that you didn’t share your code and data. This is sort of a big deal to me: I only review papers if they share data and materials. I accepted the invitation to review because I was under the impression that this would be the case. I don’t understand why you wouldn’t share. The journal reveals your names anyway and you can share data and code anonymously, like you did with the SI on the OSF. I was therefore not able to review the analysis.

Second, it would’ve been good to see a power analysis. In your preregistration, you say you will collect 600 participants for the first wave. Why this number? The references you provide there don’t tell me how you arrived at that number. What was the smallest effect size of interest you aimed to detect (Lakens et al., 2018)? Without that, it’s hard to tell whether you’re overpowered, underpowered, or properly powered.

Alternatively, I’m also happy with simply acknowledging that it’s hard to calculate power/sensitivity and you merely collected as much data as possible (Albers & Lakens, 2018), http://daniellakens.blogspot.com/2020/08/feasibility-sample-size-justification.html.

Third, I was surprised to see how quickly participants filled in the survey. Did they really take less than four minutes for the final survey? That strikes me as low given that you presented them with several scales. Did you inspect the distributions of response time and check for rushing/straightlining of survey responses? Out of experience, 5% of a sample is potentially poor quality data. (If that’s in the SI, I must’ve overlooked it and apologize. Then I’d ask the authors to include this in a footnote.)

Fourth, the exclusion criterion 3 resulted in three exclusions, but also wasn’t preregistered. Please make that clear in the paper.

Fifth, the model fitting procedure isn’t completely clear to me. You rely on SEM, yet you aggregate all scales to a mean indices. I’d argue for explicitly modelling measurement error (so a fully latent SEM) rather than pretending it’s not there (aggregating variables). If you have good reason why you didn’t go fully latent, I’m open to hearing them in your response letter. (Full disclosure: I’ve also aggregated for path models before, but have come to believe that that’s problematic.)

Sixth, your preregistration is rather vague on the analyses you’ll run. I’d acknowledge that in the manuscript. Overall, you’re extremely transparent (again: my compliments), and you ran several models to make sure you’re not cherry picking your results (and again: that’s very impressive). As you discuss constraints vs. no constraints I’d add that you do this because you didn’t restrict your researcher degree’s of freedom enough in the preregistration. I think that’d be fully transparent and current gold standard. Well, more like future gold standard, but why not start now.

Seventh, to continue the trend of transparency: You also collected other variables. I think it deserves a paragraph (or footnote) explaining what those variables are, a link to the other preregistration, to show readers that you indeed were working on separate research questions.

Eight, I think your variables are conceptually related enough to adjust your alpha to control for the family-wise error rate. You run three models times 4 for different constraints. I’m fine with not adjusting your alpha within the comparison of the four different constraint models. But between the three models, I’d argue that the outcomes are related enough that you want to guard yourself against possible false-positives. Some paths will be nonsignificant after such a correction, but results shouldn’t matter for publication; the method does, and your method is strong.

Ninth, you interpret nonsignificant effects as null effects, against which I advise caution. Absence of evidence isn’t evidence of absence, meaning just because a path isn’t significant, you can conclude that it doesn’t matter/the effect isn’t there (Greenland et al., 2016). So I’d recommend to either change statements of the absence of effects to “nonsignificant” or rely on Bayesian approaches which can provide evidence for the null (Dienes, 2019).

Tenth, I’d like to see at least a discussion of why you chose a four-month lag. You talk about the lag in the discussion, but I’d like to hear your reasoning from when you planned the study.

Discussion

First, the discussion often doesn’t go beyond mere description and summary. This point relates back to my thoughts on the introduction: Without a clear theoretical rationale in the introduction, it’s not clear what the results mean or don’t mean for self-control/mindfulness. In the discussion, you refer to Slater’s dynamic model, but without details on the model, readers won’t know how to interpret your findings in support or lack of support of that model.

Second, your falsification criteria aren’t clear. You hypothesized reciprocal effects. So under what circumstances would you say the data fully support your hypothesis, partially support it, or don’t support it? Would a reciprocal effect across two waves be enough for full support? At what point would you consider the hypothesis falsified; do all cross-lagged paths need to be nonsignificant or the majority? You’re extremely careful in the discussion, so this isn’t a lot of work to fix, merely a sentence or two more on what you would’ve considered convincing evidence.

Third, you only talk about possible positive effects of social media use in the discussion after your predictions didn’t pan out. At that point, they come a bit out of the blue, so I’d kindly ask you to discuss possible positive effects in the intro (see comment on intro).

Fourth, there are some inconsistencies in your recommendations. On line 396 you say that mindfulness training might be beneficial, but the relationship was partly reciprocal, so strengthening self-control could be just as beneficial. You acknowledge that yourself in the last sentence of the discussion.

Fifth, I disagree with you on the role of mindfulness. On line 438 you imply that mindfulness might be a mediator, but it’s not clear why this should be the case. Just because social media self-control failure predicts mindfulness and mindfulness predicts well-being at some point during the eight-month window doesn’t mean mindfulness is a mediator. You didn’t formally test mediation and there could be any number of third variables or reverse causality at work. Maybe I misunderstood the argument, so I’d ask the authors to clarify – especially because the rest of the manuscript is so careful in its conclusions.

Other

Please provide more information on the OSF when you upload the data. I kindly ask for codebooks to make it possible for readers to reproduce your analysis. In my view, computational reproducibility is a minimum requirement (Goodman et al., 2016; Hardwicke et al., 2020). In addition, R packages will certainly change in the future, which might break your code. Therefore, please at least include the output of sessionInfo() in the R script (Peikert & Brandmaier, 2019).

Random thoughts

• Page 6: The dynamic model of media effects is quite prominent, but you don’t explain it. Not all readers will be familiar with it.

• Page 7, line 138: The idea that a lot of time online will decrease offline circles has a lot of evidence against it (Antheunis et al., 2015; Dienlin et al., 2017; Przybylski & Weinstein, 2017)

• You might want to consider using omega instead of alpha (Hayes & Coutts, 2020)

• I don’t understand what “reciprocal effects might be prospectively observed” (line 390) means

Conclusion

Again, I want to express to the authors that I enjoyed reading your manuscript. My compliments on your work. I wrote a lot but that’s because I think you present strong work and as a review I’d like to help make your manuscript even stronger. I think your paper could go to print as is and would be a valuable contribution to the field, but a revision can make it a central piece of work in the discourse on self-control and social media. I look forward to seeing it in print after a revision.

If you believe some (or all) of my concerns are a result of a misunderstanding from my side, if you have any questions regarding my review, or you disagree with any of the points I raised, please feel free to contact me. I did not make confidential remarks to the editor.

I hope my suggestions were of help and I look forward to reading more of your work in the future. Until then I wish you all the best.

I always sign my reviews,

Niklas Johannes

Aagaard, J. (2020). Beyond the rhetoric of tech addiction: Why we should be discussing tech habits instead (and how). Phenomenology and the Cognitive Sciences. https://doi.org/10.1007/s11097-020-09669-z

Abendroth, A., Parry, D. A., le Roux, D. B., & Gundlach, J. (2020). An Analysis of Problematic Media Use and Technology Use Addiction Scales – What Are They Actually Assessing? In M. Hattingh, M. Matthee, H. Smuts, I. Pappas, Y. K. Dwivedi, & M. Mäntymäki (Eds.), Responsible Design, Implementation and Use of Information and Communication Technology (pp. 211–222). Springer International Publishing. https://doi.org/10.1007/978-3-030-45002-1_18

Albers, C. J., & Lakens, D. (2018). When power analyses based on pilot data are biased: Inaccurate effect size estimators and follow-up bias. Journal of Experimental Social Psychology, 74, 187–195. https://doi.org/10.17605/OSF.IO/B7Z4Q

Anderson, B. A., Laurent, P. a, & Yantis, S. (2011). Value-driven attentional capture. Proceedings of the National Academy of Sciences, 108(25), 10367–10371. https://doi.org/10.1073/pnas.1104047108

Antheunis, M. L., Vanden Abeele, M. M. P., & Kanters, S. (2015). The impact of Facebook use on micro-level social capital: A synthesis. Societies, 5(2), 399–419. https://doi.org/10.3390/soc5020399

Bayer, J. B., Campbell, S. W., & Ling, R. (2015). Connection cues: Activating the norms and habits of social connectedness. Communication Theory, 26(2), 128–149. https://doi.org/10.1111/comt.12090

Berkman, E. T., Hutcherson, C. A., Livingston, J. L., Kahn, L. E., & Inzlicht, M. (2017). Self-control as value-based choice. Current Directions in Psychological Science, 26(5), 422–428.

Dienes, Z. (2019). How Do I Know What My Theory Predicts? Advances in Methods and Practices in Psychological Science, 2515245919876960. https://doi.org/10.1177/2515245919876960

Dienlin, T., Masur, P. K., & Trepte, S. (2017). Reinforcement or displacement? The reciprocity of FTF, IM, and SNS communication and their effects on loneliness and life satisfaction. Journal of Computer-Mediated Communication, 22(2), 71–87. https://doi.org/10.1111/jcc4.12183

Domahidi, E. (2018). The associations between online media use and users’ perceived social resources: A meta-analysis. Journal of Computer-Mediated Communication, 23(4), 181–200. https://doi.org/10.1093/jcmc/zmy007

Goodman, S. N., Fanelli, D., & Ioannidis, J. P. A. (2016). What does research reproducibility mean? Science Translational Medicine, 8(341), 341ps12-341ps12. https://doi.org/10.1126/scitranslmed.aaf5027

Greenland, S., Senn, S. J., Rothman, K. J., Carlin, J. B., Poole, C., Goodman, S. N., & Altman, D. G. (2016). Statistical tests, P values, confidence intervals, and power: A guide to misinterpretations. European Journal of Epidemiology, 31(4), 337–350. https://doi.org/10.1007/s10654-016-0149-3

Hardwicke, T. E., Bohn, M., MacDonald, K., Hembacher, E., Nuijten, M. B., Peloquin, B., deMayo, B., Long, B., Yoon, E. J., & Frank, M. C. (2020). Analytic reproducibility in articles receiving open data badges at Psychological Science: An observational study. https://doi.org/10.31222/osf.io/h35wt

Hayes, A. F., & Coutts, J. J. (2020). Use Omega Rather than Cronbach’s Alpha for Estimating Reliability. But…. Communication Methods and Measures, 14(1), 1–24. https://doi.org/10.1080/19312458.2020.1718629

Hofmann, W., & Van Dillen, L. (2012). Desire: The new hot spot in self-control research. Current Directions in Psychological Science, 21(5), 317–322. https://doi.org/10.1177/0963721412453587

Inzlicht, M., & Berkman, E. (2015). Six questions for the resource model of control (and some answers). Social and Personality Psychology Compass, 9/10, 511–524. https://doi.org/10.1111/spc3.12200

Kurzban, R., Duckworth, A., Kable, J. W., & Myers, J. (2013). An opportunity cost model of subjective effort and task performance. Behavioral and Brain Sciences, 36(06), 661–679. https://doi.org/10.1017/S0140525X12003196

Lakens, D., Scheel, A. M., & Isager, P. M. (2018). Equivalence testing for psychological research: A tutorial. Advances in Methods and Practices in Psychological Science, 1(2), 259–269.

Meier, A., Domahidi, E., & Günter, E. (in press). Computer-mediated communication and mental health: A computational scoping review of an interdisciplinary field. In S. Yates & R. E. Rice (Eds.), The Oxford handbook of digital technology and society. Oxford University Press.

Panova, T., & Carbonell, X. (2018). Is smartphone addiction really an addiction? Journal of Behavioral Addictions, 7(2), 252–259. https://doi.org/10.1556/2006.7.2018.49

Peikert, A., & Brandmaier, A. M. (2019). A Reproducible Data Analysis Workflow with R Markdown, Git, Make, and Docker [Preprint]. PsyArXiv. https://doi.org/10.31234/osf.io/8xzqy

Przybylski, A. K., & Weinstein, N. (2017). A large-scale test of the Goldilocks Hypothesis: Quantifying the relations between digital-screen use and the mental well-being of adolescents. Psychological Science, 1–12. https://doi.org/10.1177/0956797616678438

Satchell, L., Fido, D., Harper, C. A., Shaw, H., Davidson, B. I., Ellis, D. A., Hart, C. M., Jalil, R., Bartoli, A. J., Kaye, L. K., Lancaster, G., & Pavetich, M. (2020). Development of an Offline-Friend Addiction Questionnaire (O-FAQ): Are most people really social addicts? [Preprint]. PsyArXiv. https://doi.org/10.31234/osf.io/7x85m

6. PLOS authors have the option to publish the peer review history of their article (what does this mean?). If published, this will include your full peer review and any attached files.

Reviewer #1: No

Reviewer #2: **Yes: **Niklas Johannes

---

## [Author Response · Author response to Decision Letter 0]

14 Mar 2021

Response to Reviewers

Response to Reviewer #1

I applaud the authors for a great paper, which I enjoyed reading, and a methodologically sound study. I have only a few recommendations to improve your work. The most important one is that you should delve deeper into the theoretical connection of the constructs of interest (beyond only quoting empirical findings), thereby enriching your theoretical background and discussion/outlook.

Response to reviewer 1 comment 1:

We appreciate the constructive and positive feedback from reviewer 1, which greatly helped us to improve this manuscript.

1. Theoretical rationale

a) Social media self-control failure (SMSCF) is an interesting construct. Nevertheless, we still know very little about it. First, I think you should make a stronger case for its relevance. Why is it important to investigate SMSCF? Why is it superior to other constructs (e.g., trait self-control, procrastination)? etc.

Response to reviewer 1 comment 1-a:

In line with the suggestion of reviewer 1, we have added the following paragraphs to make the relevance of social media self-control failure more explicit. It should be noted that we consider social media self-control failure as a different construct rather than a superior construct compared to other constructs (p. 3-4):

“Importantly, when goal-conflict occurs, social media users frequently fail to persist with these goals and turn to social media use, even though they are aware of the possible negative outcomes [6]. Social media self-control failure is often related to procrastinatory behaviors, but may also occur in situations which are not considered as typical procrastination situations (e.g., unintended checking social media while driving a car). Social media self-control is related to more general, trait-like self-control which may affect behaviors across different situations, yet focuses on the specific self-control failure induced by social media, which may happen repeatedly and frequently in a media-rich environment. Recent studies showed that social media induced self-control failure may account for 35% of the time people spent on social media [7], making them delay other important tasks, or use their time less efficiently [8, 9]. This indicates that being unable to control one’s social media use when needed might be a prevalent problem. 

The prevalence of social media self-control failure has raised concerns that frequent failure in controlling social media use may be detrimental to users’ cognitive control processes. We propose that such failure may be associated with decreased mindfulness—a general quality of consciousness which serves to maintain and sustain attention to the present experience, and to disengage from automatic thoughts, habits, and unhealthy behavior patterns [10, 11]. The recent surge in attention for mindfulness has been attributed to the rise of social media and mobile phones, and the continuous distractions they provide (e.g., [12]). Previous studies have examined the association between mindfulness and Internet use [13], problematic smartphone use [14], Facebook use [15], and online vigilance [16]. However, whether impaired mindfulness also emerges from multiple daily instances of social media self-control failure that many social media users experience is not yet known.” 

b) Second, the dynamic and reciprocal relationships are at the core of your assumptions and modelling. Theoretically, this part definitively needs more elaboration. Your assumptions comprise important deviations from Slater’s (2007, 2015). For instance, in Slater’s (2007) model, the central variables are media use as well as cognitive and behavioral variables (e.g., attitudes, behavior, identity). Your model focuses on SMSCF (which already is a cognitive effect of media use), mindfulness, and well-being. As a minimum, I would have expected some words on how Slater’s reinforcing spirals transfers to your model.

Response to reviewer 1 comment 1-b:

We agree with reviewer 1 that the reciprocal relationships described in our assumptions, guided in part by the original model of Stater’s (2007, 2015), should be more clearly articulated. Although Slater’s original model was used to examine how media exposure at time 1 influences behavior or attitudes at time 2, and vice versa, the model has also been extended to describe the reciprocal effects of specific types of media use and individual factors (e.g., media multitasking and sleep problems, see Baumgartner et al., 2017), which is why we included it in our introduction. After reading your comments (and also reviewer 2, comment 3-1), we decided to remove references to Slater from the introduction and replace it by a more specific treatment of the reciprocal relationships between our key variables. For this, we refer to our reply to reviewer 2 comment 1-2. Thus, Slater’s reinforced spiral model no longer plays a role in the argumentation leading up to our hypotheses, yet we still refer to it in the discussion (p. 33): 

“The reinforced spiral model of media effects proposes that selected exposure to certain kinds of media content could affect people’s attitude or cognition, which will increase the possibility of future exposure to the same kinds of media, and vice versa [58]. Baumgartner et al. (2017) extended the boundary of this model by examining the reciprocal relationship between specific types of media use (i.e., media multitasking) and individual difference factors (i.e., attention problems) [18].”

c) From a theoretical perspective, I’d also like to read more about the theoretical connections between SMSCF, mindfulness, and well-being. Although you provide empirical evidence, little is said in terms of why they should be related (e.g., how can SMSCF theoretically affect trait mindfulness).

Response to reviewer 1 comment 1-c:

We agree with reviewer 1 that the theoretical connections between social media self-control failure, mindfulness and wellbeing can be more clearly specified. The revised paragraphs regarding the association between social media self-control failure and mindfulness now read as follows (p. 5-6): 

“The concept of mindfulness stems from a form of meditation practice in Buddhism [21]. It is often defined as the disposition or state of being attentive to the ongoing activities or experiences in a nonjudgmental and receptive way [10, 11, 21, 22]. One of the most important components of mindfulness is the capability of acting with awareness. For instance, Bishop et al. (2014) posit that mindfulness is a two-component construct which consists of the self-regulation of attention and adopting an orientation toward one’s experiences in the present moment [22]. Baer et al. (2016) identified five components of mindfulness, two of which (i.e., observing one’s inner experience and acting with awareness) reflect the component of self-regulation of attention in Bishop’s definition [23]. Acting with awareness characterizes mindfulness as a state in which one can sustain and maintain attention to ongoing events and experiences. Mindfulness can be seen as an enhanced state of consciousness [21] which allows the observing of one’s internal or external state, that is, to “stand back and simply witness the current experience, rather than be immersed in it” [21]. Moreover, mindfulness increases feelings of vividness and awareness of one’s current experience, which helps to clarify one’s needs or interests [11], and to disengage from automatic thoughts and mind wandering [10].

Mindfulness promotes self-control through directing one’s attention in a deliberate manner [24]. Self-control is known as the ability to override automatic behavioral tendencies towards temptations that conflict with higher standards or goals [6, 25]. During this process, mindfulness could serve as a ‘detector’ of the affective cues related to these higher standards or goals [26]. For example, people with better mindfulness might be more sensitive to the affective cues related to guilty feelings when other immediate eating desire conflicts with their dieting goals. Conversely, people with lower mindfulness might be less sensitive in detecting these affective cues, which makes it more difficult in deploying self-control against the temptations. Empirical evidence confirmed this idea, showing that better self-control in general was related to higher levels of mindfulness (e.g., [27, 28]).

Regarding social media-induced self-control failure, people who fail more often to control their social media use might also show less mindfulness in their daily lives. A notable example is distraction by instant messaging [29] or notifications [30], which for many people occurs numerous times each day (an average of 64 times per day according to Pielot, Church, and De Oliveira [31]). The easy accessibility of social media creates social pressure to be always available online [32] and generates over-concern about what happens online [33]. Social media thus creates difficulties for people to maintain sufficient awareness of ongoing activities or goals. Besides, everyday use of social media is often characterized by habitual social media checking behavior, which promotes automatic use of social media [34], and which further weakens the involvement of consciousness. Together, the evidence suggests that social media self-control failure might be related to the capability to deploy attention and awareness to the present tasks or goals, which is one significant component of mindfulness.

The revised paragraphs regarding the association between social media self-control failure and wellbeing now reads as follows (p. 8-10):

“Wellbeing is often defined as the presence of positive emotions, low levels of negative emotions, and the positive judgment of one’s life in general [6, 40]. It is well-established that self-control plays an important role in wellbeing. This is because self-control prevents people from being distracted by pleasurable but unwanted desires, which benefits the goal-pursuit process [41]. Previous research has shown that people with higher trait self-control reported higher momentary happiness even as they experience the pleasurable desires [42]. This might be due to the fact that self-control over unwanted desire per se might promote the initiation of goal-pursuit behaviors [43]. This practice could lead to a less conflicting life experience, which benefits general life satisfaction. Conversely, people with lower self-control could experience that they have more conflict to deal with, and more emotional distress related to this conflict, which negatively affects their wellbeing.

Similar reasoning can be applied to self-control failure induced by social media. People who fail to control social media use are typically aware that they do so [6], which might lead to feelings of guilt or shame about their social media use [7], and in the end negatively affect their wellbeing. For example, one study found that using Facebook induced procrastination and increased students’ academic stress and strain [9]. Insufficient self-control over Internet use was found to lead to stress, anxiety and depression, particularly for those with a higher level of trait procrastination [44]. Moreover, insufficient self-control over (social) media use may also result in diminished physical activity [45] and undermine sleep quality [46], both of which are related to lower wellbeing. Overall, the evidence above implies that social media self-control failure might have negative implications for different domains of wellbeing.

However, whether decreased wellbeing conversely leads to more self-control failure in social media use is less clear. Theoretically, low levels of wellbeing might increase the risk of social media self-control failure because people may seek ways to alleviate their negative emotions. As social media can be used for mood regulation (e.g., entertainment, social contact) [47], using social media could become an attractive way to alleviate a bad mood, which challenges one’s self-control. Moreover, decreased wellbeing might also result from stress generated by goals related to study and work, because they typically require volitional effort and delaying them can create time pressure [6, 9]. The attempt to release stress thus could make social media temptation become even stronger. 

The evidence above suggests a reciprocal relationship between social media self-control failure and wellbeing. Regarding the possible reciprocal relationship between social media self-control failure and wellbeing, recent studies mainly focused on the frequency of social media use and its relationship with wellbeing. The findings were inconsistent. For instance, research based on a large-scale panel dataset showed that social media use was negatively and reciprocally related over time to teenagers’ wellbeing, although the effect sizes were trivial [48]. Another study examined the reciprocal association between use of social network sites and wellbeing. Results showed that at the between-person level, people who more often use social network sites reported lower wellbeing. However, at the within-person level, no reciprocal association was found between social network sites use and wellbeing [49]. Moreover, a meta-analysis of 124 studies showed that only certain types of social media use (e.g., online gaming) negatively impact wellbeing, while other media use (e.g., texting) positively affects wellbeing [50].”

d) Psychological well-being is misconceived. What you refer to is often understood as subjective or hedonic well-being, whereas psychological or eudaimonic well-being often relates to other theoretical concepts and operationalizations (Keyes et al., 2002; Martela & Sheldon, 2019; Ryff & Singer, 1998). Thus, the corresponding parts need clarification.

Response to reviewer 1 comment 1-d:

We agree and followed the suggestion of reviewer 1. We have now changed the label “psychological wellbeing” into “wellbeing” following the categorization of Hofmann, Reinecke, & Meier (2017). In their review, they identified two sorts of wellbeing: affective wellbeing and cognitive wellbeing (or life satisfaction), and labelled them as “wellbeing”. As we also examined both affective wellbeing (subjective vitality) and cognitive wellbeing (life satisfaction), to keep consistent with their categorization and labelling, and prevent confusion, we changed its label into “wellbeing”. We have added the citation of Hofmann et al. (2017) in the manuscript when defining wellbeing (p. 8):

“Wellbeing is often defined as the presence of positive emotions, low levels of negative emotions, and the positive judgment of one’s life in general [6, 40].”

e) I’m not fully convinced of your elaboration on stability and change. I think you need to flesh out the trait and state components of the constructs you’ve theorized and also operationalized. For instance, all the measures you’ve applied can be considered as capturing state and trait components. Thus, the random intercept cross-lagged panel model (RI-CLPM) is an appropriate way to account for this. But I’m not sure if the operationalizations are those of variables that are really susceptible to change. All of them seem to rather tap into trait-like or generalizable statements than assess a specific state. There is no particular time reference in your measures—neither in the item stems nor in the response options. An item such as “How often do you give in to a desire to use social media even though your social media use at that particular moment makes you delay other things you want or need to do?” with response options ranging from “never” to “always” doesn’t make clear if this measures a situational behavior or a more stable personality trait. As you’ve shown (Du et al., 2018), the scale has high test–retest reliability and thus seems to rather measure a trait variable. Similarly, this also holds true for the mindfulness and the life satisfaction measure. Subjective vitality could be more prone to change but this may totally depend on your instruction (e.g., rate how you feel today, the last week, the time since the last questionnaire etc.). Still, there could be a large portion of trait variance influencing the responses.

In short, the role of trait and state components is appropriately accounted for in the RI-CLPM but neither discussed theoretically nor operationalized in a suitable way.

Response to reviewer 1 comment 1-e:

We thank reviewer 1 for these insightful and helpful comments. We now discuss more specifically how trait and state components are both involved in social media self-control failure, mindfulness, and wellbeing (p. 8):

“As stated before, social media self-control failure, mindfulness and wellbeing contain both enduring trait-like individual differences, and state-like within-person variance indicating change over time. For example, a recent study found that the probability people fail to control their social media varies within persons, but the likelihood of momentary failure also differs from person to person [39]. Moreover, Du et al. (2018) found that social media self-control failure was only moderately correlated with trait-like self-control (i.e., trait self-control and depletion sensitivity), indicating that it may also capture state-like variance. The same can be said for mindfulness. At the between level, people’s trait-like, dispositional mindfulness may differ from person to person; at the within-level, their mindfulness performance can vary across time, as shown in previous research which used mindfulness training to improve people’s mindfulness performance [10]. This suggests that when exploring the reciprocal relationships of the two constructs, disentangling the trait-like individual differences and within-person change over time is necessary. To do so, we applied a random-intercept cross-lagged panel model to study the reciprocal relationships between social media self-control failure and mindfulness.” 

We agree with reviewer 1 that not specifying the time frame of the measurements might be a limitation. We now address this issue in the limitation section of the revised manuscript (p. 38):

“In addition, participants were instructed to indicate their current level of mindfulness and subjective vitality. For social media self-control failure and life satisfaction, we did not specify the time frame of the questions. Thus, some participants might have been prone to report their trait-like, general level of social media self-control failure and life satisfaction. Yet, after establishing the trait-like individual differences of these variables in the RI-CLPM, we still found the within-person effect of social media self-control failure on mindfulness, and the within-person association between life satisfaction and future social media self-control failure. Nevertheless, it could be that the instruction of the measurements reduced within-person variance in some of our measurements. Therefore, future research should take into account the time frame of the instruction regarding the measurements.”

2. Methodological soundness

Besides 1e), there’s little to moan with regard to methods, although it would have been great to inspect the analyses’ code and check for reproducability (see also 4b).

Response to reviewer 1 comment 2:

We agree and followed the suggestion of reviewer 1. We have now uploaded the anonymous dataset, r script and codebook to Open Science Framework to ensure that the results are reproducible. We now make this clear in the method section of the revised manuscript (p. 11):

“The anonymized raw datasets and syntaxes for data analysis is available online at https://osf.io/hzy8r/?view_only=0c16c08887da41eb8b08134f702a33e6.”

a) A bit more methodologically sophisticated approaches would check the construct validity of the constructs using a confirmatory factor analysis and also test for measurement invariance across measurement occasions (Brown, 2015; Geiser et al., 2015). Perhaps not essential, but could be used to show distinctness of the constructs within and across waves and comparable meaning of the measures across the three waves.

Response to reviewer 1 comment 2-a:

We thank reviewer 1 for these suggestions. First, in line with the suggestion of reviewer 1, we tested the construct validity of the scales using a confirmatory factor analysis and added the following sentences (p. 15):

“We conducted a confirmatory factor analysis to check the construct validity for each of the scale at each wave. Results showed that a one-factor model with all items had a good fit for each of the scales (See S3 File for full results).”

Second, we are aware that testing the measurement invariance is necessary, which can help compare the relationship between observed and latent variables at different times of measurement. We did not conduct this in our study, because we used the mean score as the indicator of the observed variables, and fixed its factor loading to one. In other words, the RI-CLPM we used is represented as a structural model, which is a common practice in analysing RI-CLPM (e.g., Boer, Stevens, Finkenauer, & Regina van den Eijnden, 2020; Boer, Stevens, Finkenauer, deLooze, & van den Eijnden, 2020; Baumgartner et al., 2017; Trepte and Reinecke, 2013, van der Schuur et al., 2018). Please also see our response to reviewer 2 question 5 for more information regarding this issue.

b) I’m not sure why Facebook use was measured. Moreover, in the study using the t1 data (Du et al., 2019), you reported the same items including the terms “social media” not Facebook.

Response to reviewer 1 comment 2-b:

By mistake we wrote “Facebook” while we should have written “social media”. We corrected this error, and the revised description now reads as follows (p. 15):

“Moreover, we assessed participants’ social media use with two questions: 1) “On average, approximately how many minutes per day do you spend on social media?” (1 = 10 minutes or less, 6 = 3+ hours), 2) “On average, how often do you visit social media?” (1 = Less than once a day, 6 = More than 3 times an hour).”

c) In general, I wondered why no control variables were included, at least in the unpreregistered exploratory analyses. Your own studies showed correlations with social media use, age, and gender. Wouldn’t it be useful to check their relation to the constructs of interest as well?

Response to reviewer 1 comment 2-c:

Following the suggestion of reviewer 1, in the exploratory analysis, we have now controlled for these variables. Following the procedure of adding time-invariant predictors in RI-CLPM (Mulder & Hamaker, 2020), we tested the main hypothesis after controlling for age, gender, frequency of visits on social media and time spent on social media. The revised paragraphs now read as follows (p. 20-21):

“In order to check for the robustness of the results, in an exploratory analysis we added age, gender, and social media use (both the frequency of visit on social media and time spent on social media) as control variables to the model. Specifically, the control variables were set to predict the observed variables in three waves, and its effect on each set of variables (e.g., mindfulness from time1 to time 3, social media self-control failure from time 1 to time 3) was constrained to be equal at each occasion (see Mulder & Hamaker (2020) for more information regarding the practice of including a time-invariant predictor in RI-CLPM). The full results can be found in S2 file.

Regarding the associations of the trait-like individual differences between the two variables, the results remained unchanged after controlling for age, gender and time spent on social media. Yet, adding frequency of visiting social media to the model did affect the results which in two of the four models (the models which constrained either the cross-lagged paths or both the cross-lagged and autoregressive paths) were no longer significant.”

On p 23, we added:

“Adding age, gender, frequency of visit on social media and time spent on social media as control variables did not affect any of the results.”

On p. 25, we added:

“Controlling for gender, frequency of visit on social media, and time spent on social media did not affect the results regarding the associations between the trait-like individual differences. Yet, after controlling for age, in the unconstrained model the negative association between social media self-control failure and life satisfaction became significant.

The results regarding within-person change remained the same after controlling for age and frequency of visit on social media. However, after controlling for gender, in the model which constrained the autoregressive paths to be equal, there was no longer a significant path from life satisfaction T2 to social media self-control failure T3. Also, controlling for time spent on social media rendered the paths from life satisfaction T1 to social media self-control failure T2, and from T2 to T3 were no longer significant in the model that constrained the cross-lagged paths.”

In addition, we now also discuss the methodological contribution of adding these control variables in the discussion section (p. 38):

“In additional (not preregistered) analyses, we controlled for age, gender and social media use in each of the main models. Results showed that the relationship of the trait-like individual differences between social media self-control failure and mindfulness remains almost unaffected by including the control variables, yet within-person growth of these constructs was more affected. The same pattern can be seen in the relationship between social media self-control failure and life satisfaction. The results indicate that the robustness of the results is different at the within-person level and between-person level given the influence of control variables. This further stresses that when examining the longitudinal relationships between these psychological constructs, it is necessary to apply an approach to distinguish the trait-like and state-like variances.”

3. Discussion

a) I’d like to see a more thorough discussion against the backdrop of recent findings (e.g., Liu et al., 2019; Orben et al., 2019; Stavrova & Denissen, 2020). Orben et al. (2019) is already briefly mentioned in some parts of the discussion. You could flesh out these sections.

Response to reviewer 1 comment 3-a:

Following the suggestion of reviewer 1, we now discuss these studies in the context of our research in more detail. When revising the manuscript, however, we felt that the introduction was the best place to do so. We now discuss this in more detail on p. 10-11:

“The evidence above suggests a reciprocal relationship between social media self-control failure and wellbeing. Regarding the possible reciprocal relationship between social media self-control failure and wellbeing, recent studies mainly focused on the frequency of social media use and its relationship with wellbeing. The findings were inconsistent. For instance, research based on a large-scale panel dataset showed that social media use was negatively and reciprocally related over time to teenagers’ wellbeing, although the effect sizes were trivial [48]. Another study examined the reciprocal association between use of social network sites and wellbeing. Results showed that at the between-person level, people who more often use social network sites reported lower wellbeing. However, at the within-person level, no reciprocal association was found between social network sites use and wellbeing [49]. Moreover, a meta-analysis of 124 studies showed that only certain types of social media use (e.g., online gaming) negatively impact wellbeing, while other media use (e.g., texting) positively affects wellbeing [50].

An important reason for the inconsistent findings could be that social media use per se does not necessarily result in negative outcomes, whereas problematic social media use may account for the negative outcomes, because people can use social media intensively without disturbing other daily goals. For instance, a meta-analysis showed that whereas overall social media use had no significant association with adolescents’ wellbeing, there was a significant negative association between deficient self-control over social media use and wellbeing [51]. Also, a recent longitudinal study examined the reciprocal relationship between social media use intensity and wellbeing, as well as the reciprocal relationship between problematic social media use and wellbeing. Results showed that higher levels of problematic social media use, rather than social media use intensity, were consistently related to lower life satisfaction one year later. But reversely, life satisfaction did not predict more problematic social media use [52]. ”

b) Moreover, the “social media” is a very broad term. Thus, self-control failures could refer to many different aspects of using social media features. On the one hand, given the results of Du et al. (2019), it could be interesting to discuss which features are theoretically positively related to well-being (e.g., social gratifications to satisfy belongingness or information gratifications to reduce uncertainty) but less to self-control failure, and vice versa. On the other hand, a more nuanced discussion on the potentially causal relationship between well-being and self-control failure would also help to gain insight into the relevant facets of well-being. The very broad and general Satisfaction with Life Scale (SWLS) is not necessarily directly linked to SMSCF. However, as SWLS is a cognitive indicator of subjective well-being, it could be that self-concept relevant aspects are mediating this relationship. This is just a fictitious example. As the theoretical background could be improved, so could be the discussion. More speculation about the mechanisms and conditions of the connections between SMSCF, mindfulness, life satisfaction, and vitality are necessary.

Response to reviewer 1 comment 3-b:

Following the suggestions of reviewer 1, we have now added discussion about the mechanisms between social media self-control failure and wellbeing, as well as the conditions of their connections. The revised paragraphs now read as follows (p. 34):

“The relationship between social media self-control failure and affective wellbeing may also depend on the type of gratifications expected to gain from social media use. For instance, Du et al. (2019) found that when social media use was expected to satisfy social gratifications, people were more successful in controlling their social media use [2]. However, when social media use was expected to satisfy information seeking and entertainment gratifications, people more easily failed to control their social media use. In this study, we did not distinguish between the type of social media use and its related self-control problems. Thus, it could be that self-control failures induced by specific kinds of social media use might not necessarily reduce wellbeing, which presents an interesting option to explore in future research.”

Regarding the relationship between social media self-control failure and life satisfaction, we have now added the following discussion (p. 35):

“Moreover, since life satisfaction is a product of cognitive evaluation, some self-related cognitive factors might also play an important role in determining life satisfaction. For instance, Cummins and Nistico (2002) found that the evaluation of life satisfaction can be largely biased by self-esteem and optimism. Due to these factors, people may think or report that they are happier than they really are (the “life satisfaction homeostasis”) [62]. Thus, to what extent these factors influence the link between social media self-control failure and life satisfaction at a between-level should be considered.”

c) Your limitation section starts fast-forwarding to the model constraints. Why are experimental longitudinal designs are badly needed? Why is it important to pay attention to actual and not only self-reported media use? Why is it useful to combine media use and experienced loss of self-control in a single measure? etc. These questions deserve more attention.

Response to reviewer 1 comment 3-c:

We strengthened our limitation section by elaborating on these issues. We dropped the suggestion about combining media use and experienced loss of self-control in a single measure. We deemed this suggestion to be no longer relevant given we now checked the results after controlling for social media use. The revised paragraph now reads as follows (p. 37-38):

“Yet, some limitations of the present study should be noted. First, although associations between social media self-control failure, mindfulness and subjective vitality were found using a longitudinal design, we did not experimentally manipulate these variables in order to observe their causal relationships. Thus, we cannot rule out other variables which might confound the associations between social media self-control failure, mindfulness, and wellbeing. More rigorous controlled methods could be used in future research to examine the causal relations between these variables. Second, all constructs in the study were assessed by self-report measurements, which could affect the validity in real-life scenarios. As was mentioned before, there is evidence that life satisfaction can be largely biased by one’s positive cognitions pertaining to the self, such as optimism [62]. Similarly, self-reported social media self-control failure could be biased by memory, as for many people this failure occurs frequently and may be difficult to remember correctly. Thus, using a more in-situ approach (e.g., experience sampling [16]) would provide a more objective record of real-life social media self-control failure.”

4. Open Science

Besides preregistering the present study, the present paper does not address all of proposed Open Science practices (e.g., Dienlin et al., 2020).

a. While reading the preregistration, however, I thought that it could have been beneficial to split it in two preregistrations as it has clearly been your plan from the beginning to test and publish the findings concerning your first research goal (“predictors of social media self-control failure”) in a separate paper (Du et al., 2019). Please cite this paper also on p. 8, where you refer to this publication. Moreover, I wonder whether some of those constructs you’ve examined (and found to be associated with SMSCF!) in that paper could be valuable control variables at t1 (e.g., age, online vigilance) for your present analyses. Finally, the wording of the hypotheses is not exactly the same. In the present paper, please explain why you deviate from the preregistration.

Response to reviewer 1 comment 4-a:

We are appreciated that reviewer 1 have pointed out our issues regarding our Open Science practices. 

Regarding the first research, we have now cited this paper in the manuscript. 

Second, we agree with reviewer 1 in the baseline model, some of the constructs can be controlled. We have now re-checked the results after controlling for age, gender, and social media use (please also see response to comment 2-c). However, we chose not to control for the other variables (e.g., online vigilance) because this will add too much complexity of the analysis (which is already complicated the way it is), which is also beyond the purpose of this research. 

Third, in the pre-registration, the wording of the hypotheses in the manuscript was indeed slightly different from the hypotheses in the pre-registration. But we considered they are equivalent because both build up the assumption that the individual differences of these variables will be associated cross-sectionally, and no causal statement is inferred from the two forms of hypotheses. We have added the original hypothesis to the manuscript as a footnote a. (p. 40):

“a. The original hypothesis in the pre-registration was “Social media self-control failure will negatively predict mindfulness, subjective vitality and life satisfaction cross-sectionally.”

b. Open data & material: I encourage you to upload your data and material as online supplementary material, so that both readers and reviewers can rerun analyses or check the operationalizations and implementation of the study in more detail.

Response to reviewer 1 comment 4-b:

Please refer to response to 4-c.

c. Please list all the items that were used (either in an appendix or as online supplementary material; for example, on osf.io). 

Response to reviewer 1 comment 4-b and 4-c:

We have now uploaded the anonymous dataset, r script and codebook to Open Science Framework to ensure that the results are reproducible by other researchers. Please also see response to comment 2 for the adjusted paragraph in the manuscript.

Minor issues:

– On p. 4, lines 56–58, you provide an example for a reinforced downward spiral (Orben et al., 2019). I’m not sure if this example suits your rationale because they found rather small effects and results were contingent on gender (a variable you do not include in your considerations). Moreover, the following sentence (lines 58–60) does not tell the reader why this would “provide a more complete picture.”

– p. 6: The study mentioned in Lines 113–115 (Bauer et al., 2017) is a longitudinal diary study, not a cross-sectional one.

Response to reviewer 1 comment minor issues:

We appreciate reviewer 1 for pointing out these minor issues. For the first issue, we have now removed the argument which cited Orben (2019) to underpin the downward spiral model. Instead, we now refer to this research as an example of previous inconsistent findings regarding the relationship between media use and wellbeing, (p. 10). Please see our response to reviewer 1 comment 3-a. 

For the second issue, as the whole paragraph has been adjusted, the citation is no longer fit and thus been deleted.

Reviewer #2: Dear authors,

I read your manuscript with great joy. As you know, in the literature on social media, self-control, and well-being there is a profound lack of longitudinal studies that differentiate between- and within-person effects. Also, much of the literature is opaque on data, not up-to-date with regard to Open Science practices, and often suffers from claims that simply aren’t backed by data. You make an important contribution to that literature by addressing all of these gaps. You a) preregistered your research, b) separate within- and between-person effects, c) intend to share your data and materials, and d) are modest in your conclusions. Also, you manuscript was clearly written and a pleasure to read. My compliments on your work.

During my review, I carefully examined:

• All parts of the paper

• The preregistration on the OSF

I did not:

• Review the supplementary materials carefully (lack of time)

• Review the analysis (because you didn’t provide data or code)

I will list my thoughts, concerns, and suggestions for improvement chronologically.

1. Theory Section

(1). First, the theoretical rationale wasn’t always clear to me. You define social media self-control failure and mindfulness, but it’s not clear what psychological mechanism links those two. Self-control in Psychology more and more takes the form of a motivational conflict that states that people constantly survey the value of options and contrast them with the value of their current activity/goal (Berkman et al., 2017; Kurzban et al., 2013). It’s not clear how social media self-control failure fits here. By definition, it’s a self-control process, but the mechanism behind it doesn’t become clear (yet) from the paper. You say that social media present a temptation and also speak of goal-conflict. But it’s not clear a) where that value comes from, b) in what circumstances value conflicts with another goal, c) under what circumstances people give in to that temptation. You state that this presents a “potential impairment of a psychological function that underlies intended and sustained attention”. But before you can turn to attention, it’s important to inform the reader what role attention plays in the self-control process. For example, we know that high-value options attract attention (e.g., Anderson et al., 2011). As far as I know, there’s no consensus on the role of attention within self-regulation (Hofmann & Van Dillen, 2012; Inzlicht & Berkman, 2015). It appears you use attention as an outcome of the self-regulation process: There’s a high-value option available that conflicts with my goal, so I turn my attention there. But I’m missing the explicit theoretical link here.

Also, you use attention as the link between self-control and mindfulness. In fact, you state: “Thus, mindfulness is a crucial psychological function for self-control in everyday life”. For one, this is a strong statement and I’d like to hear how mindfulness is such a central part of self-control. Conceptually, mindfulness isn’t just about self-control. As you’re aware, mindfulness describes more than just the awareness/attention component, but also evaluation etc. I’d invite the authors to elaborate here on which models of self-regulation and mindfulness they refer to. It’s unclear to me how mindfulness fits a self-regulation process other than potentially sharing a common, latent trait of attentional focus. If that’s not clear to readers, it’s not clear why you measured all facets of mindfulness and not just the attentional component.

Response to reviewer 2 comment 1-1:

We appreciate the constructive and positive feedback from reviewer 2. In line with the suggestion of reviewer 2, we have strengthened our theoretical rationale by clarifying the linkage between social media self-control failure and mindfulness. Please also see response to reviewer 1 comment 1-a.

(2). Second, most of your discussion of empirical evidence refers to concepts, not the theoretical mechanism. For example, page 6, line 102: social media use predicted burnout for those low in mindfulness. This study shares variables with yours, but I’m not sure how it supports the mechanism you propose. Mindfulness in the study is a moderator, but you use that study as support for your claim that social media self-control failure affects mindfulness directly. Another example, same page, line 108: you cite evidence that multitasking and attention were reciprocally related. I don’t see how this informs your study, unless you explain how multitasking stands in relation to social media self-control failure and how attention stands in relation to mindfulness (as a whole concept, not just the attention part). I kindly ask the authors to elaborate on their theoretical model and show how previous evidence relates to that model more clearly. Right now, most of the studies you cite just support that the concepts you study are related, but not how or why.

Response to reviewer 2 comment 1-2:

Following the suggestions of reviewer 2, we have now adjusted our formulations in the introduction. Specifically, we improved the theoretical rationale between social media self-control failure, mindfulness and wellbeing, and the empirical evidence supporting relevant arguments. We now used the study in original line 102 to support the statement that mindfulness is a buffer against the negative outcomes of media use. The more people fail to control their social media use when needed (driving a car or at work), the poorer their mindfulness may have. The adjusted paragraph now reads as follows (p. 6-7):

“Importantly, the association between social media self-control failure and mindfulness may be bidirectional. On the one hand, frequent failure in controlling social media use leads to a constantly deprived tendency to be attentive to and aware of ongoing tasks or goals. On the other hand, being unable to be attentive to and aware of ongoing tasks also increases the likelihood to be distracted by random notifications, irrelevant thoughts, and automatic checking habit of social media. Indeed, there is evidence indicating that social media use is problematic for people who showed lower levels of mindfulness. For example, a recent study found that the frequency of texting during driving was related to lower trait mindfulness through decreased self-control, which increased the risk of car accidents [35]. Another study showed that employees’ level of Facebook use was associated with lower mindfulness and subsequent burnout [15], which suggests that mindfulness may act as a buffer against the negative consequences of social media use. Conversely, lower mindfulness may also predict social media self-control failure. For example, one study found that increased mindfulness was associated with less problematic smartphone overuse [36]. Another study showed that trait mindfulness significantly decreased problematic phone use during driving [37]. Furthermore, Calvete, Gámez-Guadix, and Cortazar showed that two specific aspects of mindfulness (i.e., the extent to which one is able to observe and act with awareness) predicted deficient self-regulation of Internet use among adolescents six months later [13]. These findings suggest that lower mindfulness may indicate a higher probability of media-induced self-control failure.”

Moreover, the studies examining reciprocal relationships are now referred to support the statement that reciprocal effects are of particular interest in studies examining people’s media use and attention problems. Their work can inform our study that we will use the same way to examine the reciprocal relationships between social media self-control failure and mindfulness. The adjusted paragraph now reads as follows (p. 7):

“Although the evidence above indicates possible mutual influences between social media self-control failure and mindfulness, no study has yet examined their mutual influences over time. Thus, the possible causal relationship between the two variables remains unknown. From a dynamic perspective, in the long term, social media self-control failure might predict lower mindfulness, which in turn, might predict a higher probability of social media self-control failure in the future, and vice versa. Such reciprocal effects have been the focus of several recent studies in communication research. For instance, Baumgartner et al. (2017) found that adolescents’ media-related multitasking predicted their later attention problems, which in turn, predicted future media multitasking [18]. Boer, Stevens, Finkenauer, and Van den Eijnden (2020) found that social media use problems were reciprocally related to adolescents’ attention deficit hyperactivity disorder-symptom [38]. In a similar way, we aimed to explore the reciprocal relationships between social media self-control failure and mindfulness.”

For more revisions we made in the theory, please also see reviewer 1 comments 1-b and 1-c.

(3). Third, I ask you to consider removing all mentions of addiction from the manuscript or at least acknowledge that the concept is highly problematic and we should see it with skepticism. The literature is pretty clear that social media/internet/smartphone addiction simply is a mess of a concept/field (Aagaard, 2020; Abendroth et al., 2020; Panova & Carbonell, 2018; Satchell et al., 2020). It isn’t a clinically accepted diagnosis, so I strongly believe we should stop using it altogether until we have better evidence. I know that this is highly personal suggestion, so feel free to argue your stance in the revision letter. But I’d at least like to see an acknowledgment in the manuscript that studies use addiction, which is not a thing, but you consider it as self-control failure for reasons X and Y.

Response to reviewer 2 comment 1-3:

We share the reviewer’s concerns regarding the use of addiction as a concept in research on media use and removed all mentions of addiction.

(4). Fourth, your introduction is overall rather negative. I think it would make the paper more balanced and present a better overview of the literature for readers if you also outline the positive sides of social media (Antheunis et al., 2015; Bayer et al., 2015; Domahidi, 2018; Meier et al., in press). I think that would fit a discussion of the high value of social media because benefits and enjoyment from social media can potentially explain the high value that’s responsible for self-control failure.

Response to reviewer 2 comment 1-4:

In line with the suggestion of reviewer 2, we have now also described the benefits of social media use at the beginning of our introduction (p. 3):

“Using social media can benefit various gratifications such as social contact and entertainment [2, 3]. However, several studies have also pointed out social media use as a behavior that may disturb achieving everyday personal goals such as getting enough sleep, or devoting sufficient time to working, studying, doing house duties or engaging in sports [3-5].”

2. Method & Results

(1). First, I was disappointed to see that you didn’t share your code and data. This is sort of a big deal to me: I only review papers if they share data and materials. I accepted the invitation to review because I was under the impression that this would be the case. I don’t understand why you wouldn’t share. The journal reveals your names anyway and you can share data and code anonymously, like you did with the SI on the OSF. I was therefore not able to review the analysis.

Response to reviewer 2 comment 2-1:

Following the suggestion of reviewer 2, we have now uploaded the anonymous dataset, r script and codebook to Open Science Framework to ensure that the results are reproducible. Please also refer to reviewer 1 question 2(a).

(2). Second, it would’ve been good to see a power analysis. In your preregistration, you say you will collect 600 participants for the first wave. Why this number? The references you provide there don’t tell me how you arrived at that number. What was the smallest effect size of interest you aimed to detect (Lakens et al., 2018)? Without that, it’s hard to tell whether you’re overpowered, underpowered, or properly powered.

Alternatively, I’m also happy with simply acknowledging that it’s hard to calculate power/sensitivity and you merely collected as much data as possible (Albers & Lakens, 2018),http://daniellakens.blogspot.com/2020/08/feasibility-sample-size-justification.html.

Response to reviewer 2 comment 2-2:

We acknowledge that we did not use a power analysis to determine our sample size. We initially plan to distribute 600 surveys, which is based on the consideration that the number of observations in a SEM depends on many factors, including regressive paths, missing data, latent variables and stability of results (Wolf, Harrington, Clark, & Miller, 2013). Wolf et al. (2013) argue that SEM has advantages for examining complex associations, various types of data and comparisons across alternative models, yet these features also create difficulty to develop a generalized guideline regarding the sample size (Wolf et al., 2013). Therefore, our sample size was mainly based on studies that employed comparable sample size (e.g., Trepte & Reinecke, 2019). Moreover, we also considered the proportion of missing data after 3 waves based on our previous study employed from the same website (i.e., Prolific). Overall, taking into account comparable studies, missing data, and feasibility, we decided our initial sample size. Nevertheless, recent research also suggested feasible ways to conduct power analysis for structural equation modelling (Wang & Rhemtulla, in press, see https://yilinandrewang.shinyapps.io/pwrSEM/ for more information about the tutorial), which could be a guide for our future research. 

(3). Third, I was surprised to see how quickly participants filled in the survey. Did they really take less than four minutes for the final survey? That strikes me as low given that you presented them with several scales. Did you inspect the distributions of response time and check for rushing/straightlining of survey responses? Out of experience, 5% of a sample is potentially poor quality data. (If that’s in the SI, I must’ve overlooked it and apologize. Then I’d ask the authors to include this in a footnote.)

Response to reviewer 2 comment 2-3:

We followed the suggestion from reviewer 2. We have now checked the “speeders” of each wave of survey following Greszki, Meyer, & Schoen (2015) and also one of our previous studies. Participants who completed the survey faster than 40% of the median was excluded. We have analysed the results excluding the participants in time 1, time 2 and time 3. In total, 5 participants were excluded. No difference was found regarding the main results of our study. We also added a note of this analysis and its results as was suggested by reviewer 2 (p. 40):

“c. We also identified 5 “speeders” who completed the survey faster than 40% of the median completion time [63]. In total, 5 participants completed the survey faster than 156.4 seconds in time 1 survey, or 84.4 seconds in time 2 survey, or 78.8 seconds in time 3 survey. We re-checked the results after excluded these participants, no differences were found regarding the main results of the study.”

(4). the exclusion criterion 3 resulted in three exclusions, but also wasn’t preregistered. Please make that clear in the paper.

Response to reviewer 2 comment 2-4:

We apologize to the reviewer for this mistake. We have added this note in Table 1 that criterion (4) was not pre-registered. 

(5). Fifth, the model fitting procedure isn’t completely clear to me. You rely on SEM, yet you aggregate all scales to a mean indices. I’d argue for explicitly modelling measurement error (so a fully latent SEM) rather than pretending it’s not there (aggregating variables). If you have good reason why you didn’t go fully latent, I’m open to hearing them in your response letter. (Full disclosure: I’ve also aggregated for path models before, but have come to believe that that’s problematic.)

Response to reviewer 2 comment 2-5:

We thank reviewer 2 for pointing out this issue. In the first place, we used the mean score as the observed score and specifying its loading to 1 onto a latent variable. To our knowledge, this is common practice in analysing RI-CLPM (e.g., Boer, Stevens, Finkenauer, & Regina van den Eijnden (2020), Boer, Stevens, Finkenauer, deLooze, & van den Eijnden (2020), Baumgartner et al., (2017), Trepte and Reinecke (2013), van der Schuur et al., (2018). Thus, there is little experience in using a full latent model to analyse this model. In order to make our results more comparable to researchers conducting the same analysis, and to facilitate the understanding of the results to the readers, we prefer to keep the analysis the way we did it before. 

(6). Sixth, your preregistration is rather vague on the analyses you’ll run. I’d acknowledge that in the manuscript. Overall, you’re extremely transparent (again: my compliments), and you ran several models to make sure you’re not cherry picking your results (and again: that’s very impressive). As you discuss constraints vs. no constraints I’d add that you do this because you didn’t restrict your researcher degree’s of freedom enough in the preregistration. I think that’d be fully transparent and current gold standard. Well, more like future gold standard, but why not start now.

Response to reviewer 2 comment 2-6:

We agree with reviewer 2 that we should explain why we reported results in each constraint condition. In the limitation section of this paper, we have discussed that we did not pre-consider the constraint conditions when we pre-registered the study (p. 38).

“Importantly, we have chosen to test different models in which we varied the constraints on the different paths in the model. In our pre-registration, we did not specify how we would deal with constraints. While analyzing the data, we found different model specifications to lead to different outcomes. We have chosen to report all the results since there are arguments both pro and con applying constraints.”

(7). Seventh, to continue the trend of transparency: You also collected other variables. I think it deserves a paragraph (or footnote) explaining what those variables are, a link to the other preregistration, to show readers that you indeed were working on separate research questions.

Response to reviewer 2 comment 2-7:

First, in line with the suggestion of reviewer 2, we now added a footnote of the variables that collected in time 1 (p. 40). 

“b. The other variables collected in T1 were: immediate gratifications from social media use, habitual checking of social media, perceived ubiquity of social media, and notifications received from social media.”

Second, we agree with reviewer 2 that adding a link to the pre-registration file of the other study will increase the transparency of the study. However, as we did not separate the pre-registration of the two studies, and the pre-registration has been made public since the other study has been published, an actual link will be added after the paper was published.

(8). Eight, I think your variables are conceptually related enough to adjust your alpha to control for the family-wise error rate. You run three models times 4 for different constraints. I’m fine with not adjusting your alpha within the comparison of the four different constraint models. But between the three models, I’d argue that the outcomes are related enough that you want to guard yourself against possible false-positives. Some paths will be nonsignificant after such a correction, but results shouldn’t matter for publication; the method does, and your method is strong.

Response to reviewer 2 comment 2-8:

We thank reviewer 2 for this helpful comment. As far as we understand, the family-wise error rate is used to correct the false rejection when testing one hypothesis several times. However, for parameter estimation we used bootstrapping method and resampling 5000 times for each of the three models. Different from traditional ways of controlling family-wise error rate (e.g., Bonferroni procedure), the bootstrapping procedure accounted for the dependence structure of the p-values (Ramano & Wolf, 2008). It is also a powerful method to correct this multiple comparison error. Therefore, we believe that this issue has been tackled in our study.

(9). Ninth, you interpret nonsignificant effects as null effects, against which I advise caution. Absence of evidence isn’t evidence of absence, meaning just because a path isn’t significant, you can conclude that it doesn’t matter/the effect isn’t there (Greenland et al., 2016). So I’d recommend to either change statements of the absence of effects to “nonsignificant” or rely on Bayesian approaches which can provide evidence for the null (Dienes, 2019).

Response to reviewer 2 comment 2-9:

We agree with reviewer 2 that nonsignificant effect is not null effects. We have now adjusted the expressions in the manuscript to reduce the confusion and mis-interpretation of the readers.

(10). Tenth, I’d like to see at least a discussion of why you chose a four-month lag. You talk about the lag in the discussion, but I’d like to hear your reasoning from when you planned the study.

Response to reviewer 2 comment 2-9:

Following the suggestion of reviewer 2, we have now discussed why we chose this time lag: (p. 32)

“We set the interval between each survey at four months based on the study by Suh, Diener, and Fujita (1996), which suggests that one should have sufficient time between the measurement times in order to detect potential fluctuations in life satisfaction. Suh et al. used 3 months for their time lag [57], we chose four months between the measurements in order to avoid the time 2 survey to be in the holiday season.”

3. Discussion

(1). First, the discussion often doesn’t go beyond mere description and summary. This point relates back to my thoughts on the introduction: Without a clear theoretical rationale in the introduction, it’s not clear what the results mean or don’t mean for self-control/mindfulness. In the discussion, you refer to Slater’s dynamic model, but without details on the model, readers won’t know how to interpret your findings in support or lack of support of that model.

Response to reviewer 2 comment 3-1:

In line with the suggestion from reviewer 2, we have now strengthened our discussion regarding the spiral model and its extension. Please also see response to reviewer 1 comment 1-b.

(2). Second, your falsification criteria aren’t clear. You hypothesized reciprocal effects. So under what circumstances would you say the data fully support your hypothesis, partially support it, or don’t support it? Would a reciprocal effect across two waves be enough for full support? At what point would you consider the hypothesis falsified; do all cross-lagged paths need to be nonsignificant or the majority? You’re extremely careful in the discussion, so this isn’t a lot of work to fix, merely a sentence or two more on what you would’ve considered convincing evidence.

Response to reviewer 2 comment 3-2:

We agree with reviewer 2 that our falsification criteria regarding the hypotheses were unclear. We have now added explanations to this issue in the discussion section (p. 31): 

“It should be noted that the partial reciprocal relation neither fully supports nor rejects our hypothesis H1. We did not preregister how we would apply constraints to our model, and how we would treat differences between the results under each condition. Neither did we set clear rules for whether a reciprocal effect between two (instead of three) waves supports a reciprocal effect. Since the falsification criteria for our hypotheses were not fully detailed beforehand, the results should be interpreted with care. This also suggests that future research using RI-CLPM should include the criteria for applying constraints and the falsification criteria while designing the study.”

(3). you only talk about possible positive effects of social media use in the discussion after your predictions didn’t pan out. At that point, they come a bit out of the blue, so I’d kindly ask you to discuss possible positive effects in the intro (see comment on intro).

Response to reviewer 2 comment 3-3:

We agree and revised the introduction by starting with several positive sides of social media use. Please also refer to response to comment 1-4.

(4). Fourth, there are some inconsistencies in your recommendations. On line 396 you say that mindfulness training might be beneficial, but the relationship was partly reciprocal, so strengthening self-control could be just as beneficial. You acknowledge that yourself in the last sentence of the discussion.

Response to reviewer 2 comment 3-4:

We have now added both directions of implication regarding social media self-control failure and mindfulness. The adjusted paragraph now reads as follows (p. 33-34):

“The within-person relationship between social media self-control failure and mindfulness has implications for interventions to improve self-control in (social) media use. Elkins-Brown et al. (2017) propose that mindfulness may sensitize people to affective change when goal-conflict occurs, which benefits goal-directed behaviors and self-control. Indeed, in the clinical field, mindfulness training was demonstrated to improve healthy behaviors, such as healthy sleep habits, eating practice and physical exercise [59]. The results imply that mindfulness training may also be of help to strengthen media related self-control. Conversely, our findings suggest that improving social media related self-control might also improve mindfulness. For many users, social media distractions happen numerous times a day. Resisting social media temptations could be seen as a way to deploy attention awareness to frequently occurring situations which give rise to mindless social media behaviors, such as habitual social media checking, or social media induced mind-wandering. This practice could help to increase mindfulness in general.”

(5). Fifth, I disagree with you on the role of mindfulness. On line 438 you imply that mindfulness might be a mediator, but it’s not clear why this should be the case. Just because social media self-control failure predicts mindfulness and mindfulness predicts well-being at some point during the eight-month window doesn’t mean mindfulness is a mediator. You didn’t formally test mediation and there could be any number of third variables or reverse causality at work. Maybe I misunderstood the argument, so I’d ask the authors to clarify – especially because the rest of the manuscript is so careful in its conclusions.

Response to reviewer 2 comment 3-5:

We agree with reviewer 2 that it was too early to infer the mediating effect between the two variables. We have adjusted the articulation to make our implication more rigorous (p. 36): 

“Though no clear conclusion can be reached based on the data, the results imply an intermediate role of mindfulness between impaired social media self-control and decreased subjective vitality. Future research should explore how and to what extent mindfulness serves as a linkage between these two variables.”

4. Other

(1). Please provide more information on the OSF when you upload the data. I kindly ask for codebooks to make it possible for readers to reproduce your analysis. In my view, computational reproducibility is a minimum requirement (Goodman et al., 2016; Hardwicke et al., 2020). In addition, R packages will certainly change in the future, which might break your code. Therefore, please at least include the output of sessionInfo() in the R script (Peikert & Brandmaier, 2019).

Response to reviewer 2 comment 4-1:

We thank reviewer 2 for this useful comment. We have now uploaded the codebook to the Open Science Framework to make sure the results are reproducible. In the R script we added the code mentioned by reviewer 2 (i.e., sessionInfo).

(2). Random thoughts

1). Page 6: The dynamic model of media effects is quite prominent, but you don’t explain it. Not all readers will be familiar with it.

Response to reviewer 2 comment 4-2 (1):

We have adjusted the argumentation when explaining the dynamic mutual influences of the variables. We also further explained the spiral model of Slater’s (2016) in the discussion. Please also see response to reviewer 1 comment 1-b.

2). Page 7, line 138: The idea that a lot of time online will decrease offline circles has a lot of evidence against it (Antheunis et al., 2015; Dienlin et al., 2017; Przybylski & Weinstein, 2017)

Response to reviewer 2 comment 4-2 (2):

We agree with the suggestion of reviewer 2 that there is a lot of evidence against the assumption that online communication decreases offline circles. Therefore, we decided to remove the reference regarding this argument.

3). You might want to consider using omega instead of alpha (Hayes & Coutts, 2020)

Response to reviewer 2 comment 4-2 (3):

We have now reported Mcdonald’s omega of the scales instead of Cronbach’s Alpha. 

4). I don’t understand what “reciprocal effects might be prospectively observed” (line 390) means 

Response to reviewer 2 comment 4-2 (4):

We have now added an example to explain this statement. The adjusted paragraph now reads as follows (p. 33):

“Nevertheless, Slater et al. (2007) further proposed that given ongoing continuous mutual influence, the reciprocal effects might be prospectively observed [55]. In this case, adding more waves might increase the probability to observe the expected reciprocal effects. Therefore, future research could vary the time lag and time length of the reciprocal models.”

---

## [Decision Letter · Decision Letter 1]

23 Apr 2021

PONE-D-20-21281R1

The reciprocal relationships between social media self-control failure, mindfulness and wellbeing: A longitudinal study

PLOS ONE

Dear Dr. Du,

Thank you for submitting your manuscript to PLOS ONE. After careful consideration, we feel that it has merit but does not fully meet PLOS ONE’s publication criteria as it currently stands. Therefore, we invite you to submit a revised version of the manuscript that addresses the points raised during the review process.

Both reviewers are impressed with the way that you have handled their suggestions, and with the work that you have put into the revision and your explanation of the revisions. However, there are some remaining minor comments and suggestions that both the Reviewers and myself believe are important to address. The reviewers have again provided some thoughtful and detailed comments outlining these areas of improvement. For these reasons I am inviting a revision that addresses their comments.

We look forward to receiving your revised manuscript.

Kind regards,

Fuschia M. Sirois, PhD

Academic Editor

PLOS ONE

Journal Requirements:

Reviewers' comments:

Reviewer's Responses to Questions

**Comments to the Author**

1. If the authors have adequately addressed your comments raised in a previous round of review and you feel that this manuscript is now acceptable for publication, you may indicate that here to bypass the “Comments to the Author” section, enter your conflict of interest statement in the “Confidential to Editor” section, and submit your "Accept" recommendation.

Reviewer #1: (No Response)

Reviewer #2: (No Response)

2. Is the manuscript technically sound, and do the data support the conclusions?

Reviewer #1: Yes

Reviewer #2: Yes

3. Has the statistical analysis been performed appropriately and rigorously? 

Reviewer #1: Yes

Reviewer #2: Yes

4. Have the authors made all data underlying the findings in their manuscript fully available?

Reviewer #1: Yes

Reviewer #2: Yes

5. Is the manuscript presented in an intelligible fashion and written in standard English?

Reviewer #1: Yes

Reviewer #2: Yes

6. Review Comments to the Author

Reviewer #1: I already liked the first version of the manuscript and I think you did an outstanding job on revising the manuscript, carrying out additional helpful analyses, and crafting the response letter in a generally convincing and well-organized way.

I was convinced by most of your answers. Nevertheless, some minor clarifications would help further improve the paper.

1. Regarding my previous Point 2, thanks for uploading the codebook, the data, and the R code. I ran the code and it generally worked well (but see 1a and 1b for necessary modifications).

a) To foster reproducibility, it would help to include some instructions or lines of code how to load the data before running the code, for instance, in a readme file or, even better, directly within the code file.

b) Please insert that the tidyverse (or similar) package needs to be loaded. Otherwise, the %>% cannot be interpreted properly.

c) The sessionInfo() makes only sense if its output is included in a static file (e.g., a html output of an RMarkdown document). Only then, it will include the necessary info to reproduce the specific environment of your analyses. Otherwise, it will just produce the info of the session the reproducer ran.

d) To view your analyses and results, a static output file (e.g., a html output of an RMarkdown document) may be of great benefit to the readers as well, because they don’t have to carry out the cumbersome bootstrapping procedures for all the models but can just look up what you’ve done and what the output looks like.

2. Ad 2b), it’s great that you’ve conducted additional analyses including control variables and integrated the new findings. Here, three minor issues arise:

a) You should cite your own findings as a reason why you’ve included exactly these control variables (Du et al., 2019).

b) If I’m not mistaken, Mulder and Hamaker (2020) is not yet listed in the references of your paper.

c) Compared to the paper’s tables, the S2 Appendix is a bit awkward to read. I recommend a more straightforward depiction of the relevant findings to make it easier for the readers to compare the paths across models.

3. The only point where I disagree with you concerns 2a and Reviewer 2’s fifth point.

a) I don’t think that referring to “common practice” is a sound argument for not moving into a latent variables framework. On the same basis, you could have just carried out a traditional cross-lagged model. You would have found even more studies doing so but that not implies that this is the best or appropriate way to do it.

b) I also don’t buy that there’s little practice in moving to a latent framework. You’ve already drawn on Mulder and Hamaker (2020) to include the control variables. They also wrote a section on the multiple-indicator RI-CLPM. Ironically, the “common” RI-CLPM is a special case of the more general multiple-indicator RI-CPLM and both are closely connected to latent growth curves with structured residuals (Thomas et al., 2021; Usami et al., 2019; Zyphur, Allison, et al., 2019; Zyphur, Voelkle, et al., 2019).

c) Similar to Reviewer 2, please don’t get me wrong: I don’t urge you to move to a latent variables framework if you have good reasons to not do so. From my perspective, however, the reasons you’ve mentioned thus far are not enough to convince me. Moreover, the CFAs reported in S3 rather obscure than illuminate the factorial validity of the constructs. For instance, for three-item-single-factor models you have to impose constraints to make them identifiable. I recommend running multi-construct latent state–trait models (Eid et al., 1994; Geiser, 2020; Steyer et al., 2015) or, at least, CFAs including all latent constructs of interest at the same measurement occasion to examine factorial structure and potential cross-loadings.

References

Du, J., Kerkhof, P., & van Koningsbruggen, G. M. (2019). Predictors of social media self-control failure: Immediate gratifications, habitual checking, ubiquity, and notifications. Cyberpsychology, Behavior and Social Networking, 22(7), 477–485. https://doi.org/10.1089/cyber.2018.0730

Eid, M., Notz, P., Steyer, R., & Schwenkmezger, P. (1994). Validating scales for the assessment of mood level and variability by latent state-trait analyses. Personality and Individual Differences, 16(1), 63–76. https://doi.org/10.1016/0191-8869(94)90111-2

Geiser, C. (2020). Longitudinal structural equation modeling with Mplus: A latent state–trait perspective. Guilford Press.

Mulder, J. D., & Hamaker, E. L. (2020). Three extensions of the random intercept cross-lagged panel model. Structural Equation Modeling: A Multidisciplinary Journal. Advance online publication. https://doi.org/10.1080/10705511.2020.1784738

Steyer, R., Mayer, A., Geiser, C., & Cole, D. A. (2015). A theory of states and traits—Revised. Annual Review of Clinical Psychology, 11, 71–98. https://doi.org/10.1146/annurev-clinpsy-032813-153719

Thomas, F., Shehata, A., Otto, L. P., Möller, J., & Prestele, E. (2021). How to capture reciprocal communication dynamics: Comparing longitudinal statistical approaches in order to analyze within- and between-person effects. Journal of Communication. Advance online publication. https://doi.org/10.1093/joc/jqab003

Usami, S., Murayama, K., & Hamaker, E. L. (2019). A unified framework of longitudinal models to examine reciprocal relations. Psychological Methods, 24(5), 637–657. https://doi.org/10.1037/met0000210

Zyphur, M. J., Allison, P. d., Tay, L., Voelkle, M. C., Preacher, K. J., Zhang, Z., Hamaker, E. L., Shamsollahi, A., Pierides, D. C., Koval, P., & Diener, E. (2019). From data to causes I: Building a general cross-lagged panel model (GCLM). Organizational Research Methods, 1094428119847278. https://doi.org/10.1177/1094428119847278

Zyphur, M. J., Voelkle, M. C., Tay, L., Allison, P. d., Preacher, K. J., Zhang, Z., Hamaker, E. L., Shamsollahi, A., Pierides, D. C., Koval, P., & Diener, E. (2019). From data to causes II: Comparing approaches to panel data analysis. Organizational Research Methods, 1094428119847280. https://doi.org/10.1177/1094428119847280

Reviewer #2: Dear authors,

Thank you for the opportunity to read this manuscript again. I liked the original admission and am impressed by the work you invested into the revision. The paper has become even more coherent and you addressed all concerns that I had (and, from my reading, the other reviewer had). My compliments. I think your work will make a valuable contribution to the literature and I look forward to seeing it in print.

Below just a couple of random thoughts that I DON’T want you to address in a second round, but wanted to share anyway:

• “Empirical evidence confirmed” � I’d prefer “support”

• One more sentence on the possible mechanism self-control failure � lower mindfulness would’ve been helpful. Something like: “Repeatedly failing to respond to affective cues, thereby experiencing self-control failure, might further decrease people’s sensitivity to those cues over time. In other words, self-control failure � lower mindfulness”. Just a suggestion for future papers, no need to implement here. Actually, you have a similar sentence in the discussion.

• Really strong justification for distinction of trait vs. state.

• Exemplary transparency on reporting the first study based on these data.

• I’m not convinced bootstrapping solves the family-wise error rate. But I’m not a statistician, so I need to read up on this.

• Again: Exemplary treatment of the falsification criteria.

• I’m personally skeptical of the mindfulness interventions (see https://doi.org/10.1177/1745691617709589), but not sure that criticism affects your conclusions about mindfulness treatments in the discussion.

A couple of thoughts (independent of the content of the paper) on reproducibility, which can be addressed within half an hour:

• I tried reproducing your analysis: There’s no “DurationT1” variable in the data set.

• You use dplyr pipes, but don’t load any tidyverse packages.

• It would be ideal if you tried running the entire script from beginning to end on a new machine to check whether results can be reproduced.

• The output of sessionInfo() as a text file should ideally be added to the project for reproducibility.

• I had to request access to the OSF project – can you remove me again as a collaborator?

One more time: My compliments and congratulations on your work!

Niklas Johannes

7. PLOS authors have the option to publish the peer review history of their article (what does this mean?). If published, this will include your full peer review and any attached files.

Reviewer #1: **Yes: **Frank M. Schneider

Reviewer #2: **Yes: **Niklas Johannes

---

## [Author Response · Author response to Decision Letter 1]

28 Jun 2021

Reviewer #1: I already liked the first version of the manuscript and I think you did an outstanding job on revising the manuscript, carrying out additional helpful analyses, and crafting the response letter in a generally convincing and well-organized way.

Response to reviewer 1: 

We would like to thank reviewer 1 for the positive feedback on our manuscript. We are grateful for the feedback because we feel that this improved the quality of the manuscript.

I was convinced by most of your answers. Nevertheless, some minor clarifications would help further improve the paper.

1. Regarding my previous Point 2, thanks for uploading the codebook, the data, and the R code. I ran the code and it generally worked well (but see 1a and 1b for necessary modifications).

a) To foster reproducibility, it would help to include some instructions or lines of code how to load the data before running the code, for instance, in a readme file or, even better, directly within the code file.

Response to reviewer 1 comment 1-a: 

Following the suggestion of reviewer 1, we have added a line regarding how to locate the original data:

“#Read data, between the brackets should be the local location of the raw data on your device

merge <- read.csv("~/Desktop/PlosOne/Revision/7 Data analysis (study 2)/20190712 Data analysis/20190712/merge.csv")”

b) Please insert that the tidyverse (or similar) package needs to be loaded. Otherwise, the %>% cannot be interpreted properly.

Response to reviewer 1 comment 1-b: 

Thank you for pointing this out. We have now added the dplyr package in the R script: “library(dplyr)”

c) The sessionInfo() makes only sense if its output is included in a static file (e.g., a html output of an RMarkdown document). Only then, it will include the necessary info to reproduce the specific environment of your analyses. Otherwise, it will just produce the info of the session the reproducer ran.

Response to reviewer 1 comment 1-c: 

To solve this problem, we now directly copied the output information of the sessioninfo() at the beginning of the R script. We also added the sessioninfo output in the Rmarkdown html output.

d) To view your analyses and results, a static output file (e.g., a html output of an RMarkdown document) may be of great benefit to the readers as well, because they don’t have to carry out the cumbersome bootstrapping procedures for all the models but can just look up what you’ve done and what the output looks like.

Response to reviewer 1 comment 1-d:

Thank you for this helpful advice. We have now created an RMarkdown document in html version to store all scripts and results. The file can be found at the Open Science Framework https://osf.io/hzy8r/ We refer to this file on p. 11 in the revised version of our manuscript.

2. Ad 2b), it’s great that you’ve conducted additional analyses including control variables and integrated the new findings. Here, three minor issues arise:

a) You should cite your own findings as a reason why you’ve included exactly these control variables (Du et al., 2019).

Response to reviewer 1 comment 2-a: 

Following the suggestion of the reviewer, we now refer to two previous studies that showed that age, sex and social media use were related to social media self-control failure as a reason to include these as control variables. The revised paragraph now reads as follows (pp. 20):

“In order to check for the robustness of the results, in an exploratory analysis we added age, gender, and social media use (both the frequency of visit on social media and time spent on social media) as control variables to the model. The control variables were selected based on the findings in other studies, which showed that age was positively related to social media self-control failure, social media use was negatively related to social media self-control failure, and female participants tend to report higher levels of social media self-control failure than male participants [2][7].”

b) If I’m not mistaken, Mulder and Hamaker (2020) is not yet listed in the references of your paper.

Response to reviewer 1 comment 2-b:

Thank you for pointing this out; we now added Mulder and Hamaker (2020) to the references.

c) Compared to the paper’s tables, the S2 Appendix is a bit awkward to read. I recommend a more straightforward depiction of the relevant findings to make it easier for the readers to compare the paths across models.

Response to reviewer 1 comment 2-c: 

We agree with reviewer 1 that we could improve the readability of the results in S2. Thus, to compare the paths across different models with each of the original model, we described the relevant findings in paragraphs in the manuscript, and also mentioned S2 file as a reference. We now also presented the results in tables to make it easier for the readers (see S2 file for more information). The paragraphs are listed below:

SMSCF-Mindfulness (pp. 21):

“Regarding the associations of the trait-like individual differences between the two variables, the results remained unchanged after controlling for age, gender and time spent on social media. Yet, adding frequency of visiting social media to the model did affect the results which in one the four models (the models which constrained the cross-lagged and autoregressive paths) were no longer significant.

Regarding the within-person change of the two variables, adding age resulted in a full reciprocal effect between social media self-control failure and mindfulness in the model which constrained both autoregressive and cross-lagged paths to be equal. Social media self-control failure predicted lower mindfulness and vice versa. Yet, after controlling for gender or frequency of visit, the prediction from social media self-control failure T1 to mindfulness T2 were no longer significant in the unconstraint model. After controlling for time spent on social media, the cross-lagged paths were no longer significant.”

SMSCF-Subjective Vitality (pp. 23): 

“Controlling for adding age, gender, frequency of visit on social media and time spent on social media as control variables did not affect any of the results.”

SMSCF-Life satisfaction (p. 25):

“Controlling for age, gender, frequency of visit on social media, and time spent on social media did not affect the results regarding the associations between the trait-like individual differences.

The results regarding within-person change remained the same after controlling for age and frequency of visit on social media. However, after controlling for gender, in the model which constrained the autoregressive paths to be equal, there was no longer a significant path from life satisfaction T2 to social media self-control failure T3. Also, controlling for time spent on social media rendered the paths from life satisfaction T1 to social media self-control failure T2, and from T2 to T3 were no longer significant in the model that constrained the cross-lagged paths.”

3. The only point where I disagree with you concerns 2a and Reviewer 2’s fifth point.

a) I don’t think that referring to “common practice” is a sound argument for not moving into a latent variables framework. On the same basis, you could have just carried out a traditional cross-lagged model. You would have found even more studies doing so but that not implies that this is the best or appropriate way to do it.

Response to reviewer 1 comment 3-a: 

Referring to common practice may indeed not be the best argument. However, the fact that using latent variables has only recently been introduced in RI-CLPM makes for a situation where solutions for problems you may run into have not yet been provided. We have tried a model with latent variables, following the 4-step process suggested by Mulder and Hamaker (2020). However, the results showed estimation problems during each step. For instance, regarding the model of social media self-control failure and mindfulness, the configural model showed that the model converged, but not all elements of the gradient are zero; The weak factorial invariance model showed that the covariance matrix of latent variables is not positive definite; The strong factorial invariance model and the strong factorial invariance model with factor loadings equal to the within-person factor loadings showed that the variance-covariance matrix of the estimated parameters was not positive definite. Due to this issue, some regression paths were not successfully estimated in each of the steps. 

Considering the high number of items (15 items in total) in the Mindfulness Attention Awareness scale, we assessed whether the estimation problems were due to problems in the scale by testing a model using only the first 4 items to estimate the model. This model was successfully estimated, indicating that part of the problems were indeed due to the MAAS. This made us consider item parceling as a solution, however in the RI-CLPM this creates issues in the part of the model where we specify the random intercepts for each indicator separately (and in other parts as well).

In the other models we also run into estimation issues which we don’t know how to properly solve. Therefore, we feel that using latent variables brings us into uncharted territory. Solutions may exist, but given how recently latent variable RI-CLPM has been introduced, we cannot rely on the robustness and correctness of such solutions. Therefore, we prefer to hold on to using mean scores as indicators in our models. 

To discuss the issue raised by Reviewer 1, we have added the following text to the discussion (pp. 39):

“Besides, the current research is limited by using mean score of each variable to fit the RI-CLPM, based on the rather strong assumption of no measurement error. However, this could bias lagged-parameter estimates, which weakens the explanatory power of the findings [55]. We also tried the latent variable model in analyzing the data. Following Hamaker & Mulder (2020), estimating the latent model is a 4-step process which helps build measurement invariance. However, results showed estimation problems during each step, which made us decide to hold on to using mean scores as indicators in our models. Future research could use multiple indicators of each variable to fit the RI-CLPM, or use alternative reciprocal models that account for measurement error, such as the stable trait autoregressive trait and state model (see Usami et al. (2019) for a review of different types of reciprocal models) [64].”

b) I also don’t buy that there’s little practice in moving to a latent framework. You’ve already drawn on Mulder and Hamaker (2020) to include the control variables. They also wrote a section on the multiple-indicator RI-CLPM. Ironically, the “common” RI-CLPM is a special case of the more general multiple-indicator RI-CPLM and both are closely connected to latent growth curves with structured residuals (Thomas et al., 2021; Usami et al., 2019; Zyphur, Allison, et al., 2019; Zyphur, Voelkle, et al., 2019).

Response to reviewer 1 comment 3-b: 

Please see our response to 3-a. 

c) Similar to Reviewer 2, please don’t get me wrong: I don’t urge you to move to a latent variables framework if you have good reasons to not do so. From my perspective, however, the reasons you’ve mentioned thus far are not enough to convince me. Moreover, the CFAs reported in S3 rather obscure than illuminate the factorial validity of the constructs. For instance, for three-item-single-factor models you have to impose constraints to make them identifiable. I recommend running multi-construct latent state–trait models (Eid et al., 1994; Geiser, 2020; Steyer et al., 2015) or, at least, CFAs including all latent constructs of interest at the same measurement occasion to examine factorial structure and potential cross-loadings.

Response to reviewer 1 comment 3-c: 

Regarding the discussion about latent RI-CLPM, please see our response to 3-a. Regarding the CFA results, following the suggestion of Reviewer 1, we have now included all latent constructs of interest at each time in the same measurement occasion to examine the factorial structure. The results are reported in the S3 file.

Reviewer #2: Dear authors,

Thank you for the opportunity to read this manuscript again. I liked the original admission and am impressed by the work you invested into the revision. The paper has become even more coherent and you addressed all concerns that I had (and, from my reading, the other reviewer had). My compliments. I think your work will make a valuable contribution to the literature and I look forward to seeing it in print.

1 Below just a couple of random thoughts that I DON’T want you to address in a second round, but wanted to share anyway:

• “Empirical evidence confirmed” � I’d prefer “support”

• One more sentence on the possible mechanism self-control failure � lower mindfulness would’ve been helpful. Something like: “Repeatedly failing to respond to affective cues, thereby experiencing self-control failure, might fur[=ther decrease people’s sensitivity to those cues over time. In other words, self-control failure � lower mindfulness”. Just a suggestion for future papers, no need to implement here. Actually, you have a similar sentence in the discussion.

• Really strong justification for distinction of trait vs. state.

• Exemplary transparency on reporting the first study based on these data.

• I’m not convinced bootstrapping solves the family-wise error rate. But I’m not a statistician, so I need to read up on this.

• Again: Exemplary treatment of the falsification criteria.

• I’m personally skeptical of the mindfulness interventions (see https://doi.org/10.1177/1745691617709589), but not sure that criticism affects your conclusions about mindfulness treatments in the discussion.

Response to reviewer 2 comment 1:

We thank reviewer 2 for the positive feedback and the suggestions. Following the suggestion of reviewer 2, we have changed the word “Empirical evidence confirmed” into “Empirical evidence supported”, and added one sentence to further explain the relationship between social media self-control failure and mindfulness. We thank the reviewer for sharing his thoughts regarding the other issues and we will keep these in mind for future papers.

2. A couple of thoughts (independent of the content of the paper) on reproducibility, which can be addressed within half an hour:

1)• I tried reproducing your analysis: There’s no “DurationT1” variable in the data set.

2)• You use dplyr pipes, but don’t load any tidyverse packages.

3)• It would be ideal if you tried running the entire script from beginning to end on a new machine to check whether results can be reproduced.

4)• The output of sessionInfo() as a text file should ideally be added to the project for reproducibility.

5)• I had to request access to the OSF project – can you remove me again as a collaborator?

One more time: My compliments and congratulations on your work!

Niklas Johannes

Response to reviewer 2 comment 2-1:

Regarding the variable “DurationT1”, this variable name could be generated after changing the name of the raw data (the original variable name in the raw data is “Duration_in_seconds”). The corresponding R code is: colnames(merge)[colnames(merge)=="Duration__in_seconds"] <- "DurationT1". 

Response to reviewer 2 comment 2-2:

We have also added a line to load dplyr package. Please also refer to reviewer 1 response 1-b, 1-c and 1-d.

Response to reviewer 2 comment 2-3 and 2-4:

We have re-run all scripts, and also created a Rmarkdown html output file, the script, as well as all results are now stored in this output file, including the sessioninfo. 

Response to reviewer 2 comment 2-5:

We have now removed you from the collaborator list.

---

## [Editor Report · Decision Letter 2]

22 Jul 2021

The reciprocal relationships between social media self-control failure, mindfulness and wellbeing: A longitudinal study

PONE-D-20-21281R2

Dear Dr. Du,

We’re pleased to inform you that your manuscript has been judged scientifically suitable for publication and will be formally accepted for publication once it meets all outstanding technical requirements.

Kind regards,

Fuschia M. Sirois, PhD

Academic Editor

PLOS ONE
---

## [Editor Report · Acceptance letter]

27 Jul 2021

PONE-D-20-21281R2 

The reciprocal relationships between social media self-control failure, mindfulness and wellbeing: A longitudinal study 

Dear Dr. Du:

I'm pleased to inform you that your manuscript has been deemed suitable for publication in PLOS ONE. Congratulations! Your manuscript is now with our production department. 

Kind regards, 

on behalf of

Dr. Fuschia M. Sirois 

Academic Editor

PLOS ONE